# Getting a CLUE: A Method for Explaining Uncertainty Estimates

**Javier Antorán**
University of Cambridge
ja666@cam.ac.uk

**Umang Bhatt**
University of Cambridge
usb20@cam.ac.uk

**Tameem Adel**
University of Cambridge
University of Liverpool
tah47@cam.ac.uk

**Adrian Weller**
University of Cambridge
The Alan Turing Institute
aw665@cam.ac.uk

**José Miguel Hernández-Lobato**
University of Cambridge
The Alan Turing Institute
jmh233@cam.ac.uk

## Abstract

Both uncertainty estimation and interpretability are important factors for trustworthy machine learning systems. However, there is little work at the intersection of these two areas. We address this gap by proposing a novel method for interpreting uncertainty estimates from differentiable probabilistic models, like Bayesian Neural Networks (BNNs). Our method, Counterfactual Latent Uncertainty Explanations (CLUE), indicates how to change an input, while keeping it on the data manifold, such that a BNN becomes more confident about the input's prediction. We validate CLUE through 1) a novel framework for evaluating counterfactual explanations of uncertainty, 2) a series of ablation experiments, and 3) a user study. Our experiments show that CLUE outperforms baselines and enables practitioners to better understand which input patterns are responsible for predictive uncertainty.

## 1 Introduction

There is growing interest in probabilistic machine learning models, which aim to provide reliable estimates of uncertainty about their predictions (MacKay, 1992). These estimates are helpful in high-stakes applications such as predicting loan defaults or recidivism, or in work towards autonomous vehicles. Well-calibrated uncertainty can be as important as making accurate predictions, leading to increased robustness of automated decision-making systems and helping prevent systems from behaving erratically for out-of-distribution (OOD) test points. In practice, predictive uncertainty conveys skepticism about a model's output. However, its utility need not stop there: we posit predictive uncertainty could be rendered more useful and actionable if it were expressed in terms of model inputs, answering the question: *"Which input patterns lead my prediction to be uncertain?"*

Understanding which input features are responsible for predictive uncertainty can help practitioners learn in which regions the training data is sparse. For example, when training a loan default predictor, a data scientist (i.e., practitioner) can identify sub-groups (by age, gender, race, etc.) under-represented in the training data. Collecting more data from these groups, and thus further constraining their model's parameters, could lead to accurate predictions for a broader range of clients. In a clinical scenario, a doctor (i.e., domain expert) can use an automated decision-making system to assess whether a patient should receive a treatment. In the case of high uncertainty, the system would suggest that the doctor should not rely on its output. If uncertainty were explained in terms of which features the model finds anomalous, the doctor could appropriately direct their attention.

While explaining predictions from deep models has become a burgeoning field (Montavon et al., 2018; Bhatt et al., 2020b), there has been relatively little research on explaining what leads to neural networks' predictive uncertainty. In this work, we introduce Counterfactual Latent Uncertainty Explanations (CLUE), to our knowledge, the first approach to shed light on the subset of input space features that are responsible for uncertainty in probabilistic models. Specifically, we focus on explaining Bayesian Neural Networks (BNNs). We refer to the explanations given by our method as

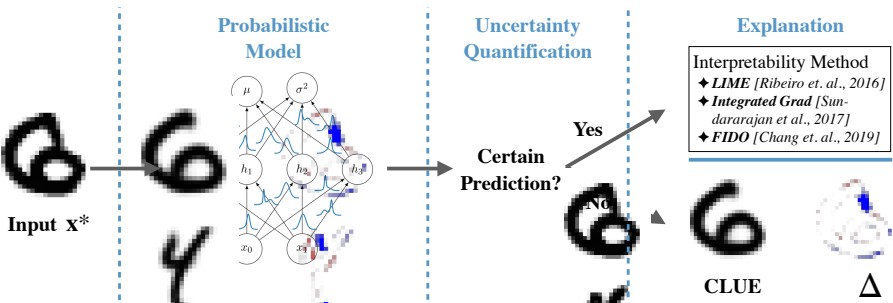

Figure 1: Workflow for automated decision making with transparency. Our probabilistic classifier produces a distribution over outputs. In cases of high uncertainty, CLUE allows us to identify features which are responsible for class ambiguity in the input (denoted by $\Delta$ and highlighted in dark blue). Otherwise, we resort to existing feature importance approaches to explain certain decisions.

CLUEs. CLUEs try to answer the question: *"What is the smallest change that could be made to an input, while keeping it in distribution, so that our model becomes certain in its decision for said input?"* CLUEs can be generated for tabular and image data on both classification and regression tasks.

An application of CLUE is to improve transparency in the real-world deployment of a probabilistic model, such as a BNN, by complementing existing approaches to model interpretability (Ribeiro et al., 2016; Sundararajan et al., 2017; Chang et al., 2019). When the BNN is confident in its prediction, practitioners can generate an explanation via earlier feature importance techniques. When the BNN is uncertain, its prediction may well be wrong. This potentially wrong prediction could be the result of factors not related to the actual patterns present in the input data, e.g. parameter initialization, randomness in mini-batch construction, etc. An explanation of an uncertain prediction will be disproportionately affected by these factors. Indeed, recent work on feature attribution touches on the unreliability of saliency maps when test points are OOD (Adebayo et al., 2020). Therefore, when the BNN is uncertain, it makes sense to provide an explanation of why the BNN is uncertain (i.e., CLUE) instead of an explanation of the BNN's prediction. This is illustrated in Figure 1. Our code is at: github.com/cambridge-mlg/CLUE. We highlight the following contributions:

- We introduce CLUE, an approach that finds counterfactual explanations of uncertainty in input space, by searching in the latent space of a deep generative model (DGM). We put forth an algorithm for generating CLUEs and show how CLUEs are best displayed.

- We propose a computationally grounded approach for evaluating counterfactual explanations of uncertainty. It leverages a separate conditional DGM as a synthetic data generator, allowing us to quantify how well explanations reflect the true generative process of the data.

- We evaluate CLUE quantitatively through comparison to baseline approaches under the above framework and through ablative analysis. We also perform a user study, showing that CLUEs allow practitioners to predict on which new inputs a BNN will be uncertain.

## 2 PRELIMINARIES

### 2.1 UNCERTAINTY IN BNNS

Given a dataset $\mathcal{D} = \{\mathbf{x}^{(n)}, \mathbf{y}^{(n)}\}_{n=1}^{N}$, a prior on our model's weights $p(\mathbf{w})$, and a likelihood function $p(\mathcal{D}|\mathbf{w}) = \prod_{n=1}^{N} p(\mathbf{y}^{(n)}|\mathbf{x}^{(n)}, \mathbf{w})$, the posterior distribution over the predictor's parameters $p(\mathbf{w}|\mathcal{D}) \propto p(\mathcal{D}|\mathbf{w})p(\mathbf{w})$ encodes our uncertainty about what value $\mathbf{w}$ should take. Through marginalization, this parameter uncertainty is translated into predictive uncertainty, yielding reliable error bounds and preventing overfitting:

$$p(\mathbf{y}^*|\mathbf{x}^*, \mathcal{D}) = \int p(\mathbf{y}^*|\mathbf{x}^*, \mathbf{w})p(\mathbf{w}|\mathcal{D}) \, d\mathbf{w}. \tag{1}$$

For BNNs, both the posterior over parameters and predictive distribution (1) are intractable. Fortunately, there is a rich literature concerning approximations to these objects (MacKay, 1992; Hernández-Lobato & Adams, 2015; Gal, 2016). In this work, we use scale-adapted Stochastic Gradient Hamiltonian Monte Carlo (SG-HMC) (Springenberg et al., 2016). For regression, we use

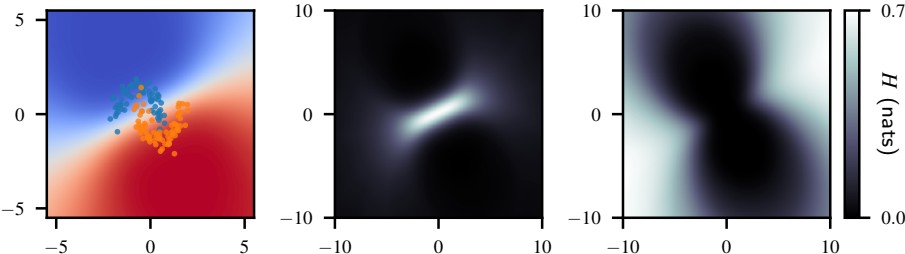

Figure 2: Left: Training points and predictive distribution for variational Bayesian Logistic Regression on the Moons dataset. Center: Aleatoric entropy $H_a$ matches regions of class non-separability. Right: Epistemic entropy $H_e$ grows away from the data. Both uncertainties are detailed in Appendix B.2.

heteroscedastic Gaussian likelihood functions, quantifying uncertainty using their standard deviation, $\sigma(\mathbf{y}|\mathbf{x})$. For classification, we take the entropy $H(\mathbf{y}|\mathbf{x})$ of categorical distributions as uncertainty. Details are given in Appendix B. Hereafter, we use $\mathcal{H}$ to refer to any uncertainty metric, be it $\sigma$ or $H$.

Predictive uncertainty can be separated into two components, as shown in Figure 2. Each conveys different information to practitioners (Depeweg, 2019). Irreducible or *aleatoric uncertainty* is caused by inherent noise in the generative process of the data, usually manifesting as class overlap. Model or *epistemic uncertainty* represents our lack of knowledge about $\mathbf{w}$. Stemming from a model being under-specified by the data, epistemic uncertainty arises when we query points off the training manifold. Capturing model uncertainty is the main advantage of BNNs over regular NNs. It enables the former to be used for uncertainty aware tasks, such as OOD detection (Daxberger & Hernández-Lobato, 2019), continual learning (Nguyen et al., 2018), active learning (Depeweg et al., 2018), and Bayesian optimization (Springenberg et al., 2016).

## 2.2 Uncertainty Sensitivity Analysis

To the best of our knowledge, the only existing method for interpreting uncertainty estimates is Uncertainty Sensitivity Analysis (Depeweg et al., 2017). This method quantifies the global importance of an input dimension to a chosen metric of uncertainty $\mathcal{H}$ using a sum of linear approximations centered at each test point:

$$I_i = \frac{1}{|\mathcal{D}_{\text{test}}|} \sum_{n=1}^{|\mathcal{D}_{\text{test}}|} \left| \frac{\partial \mathcal{H}(\mathbf{y}_n|\mathbf{x}_n)}{\partial x_{n,i}} \right|. \tag{2}$$

As discussed by Rudin (2019), linear explanations of non-linear models, such as BNNs, can be misleading. Even generalized linear models, which are often considered to be "inherently interpretable," like logistic regression, produce non-linear uncertainty estimates in input space. This can be seen in Figure 2. Furthermore, high-dimensional input spaces limit the actionability of these explanations, as $\nabla_{\mathbf{x}} \mathcal{H}$ will likely not point in the direction of the data manifold. In Figure 3 and Appendix D, we show how this can result in sensitivity analysis generating meaningless explanations.

Our method, CLUE, leverages the latent space of a DGM to avoid working with high-dimensional input spaces and to ensure explanations are in-distribution. CLUE does not rely on crude linear approximations. The counterfactual nature of CLUE guarantees explanations have tangible meaning.

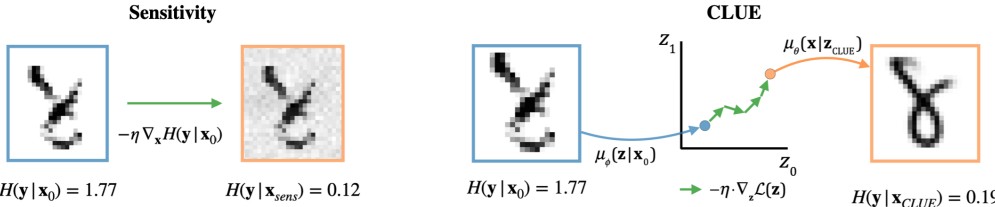

Figure 3: Left: Taking a step in the direction of maximum sensitivity leads to a seemingly noisy input configuration for which $H$ is small. Right: Minimizing CLUE's uncertainty-based objective in terms of a DGM's latent variable $\mathbf{z}$ produces a plausible digit with a corrected lower portion.

## 2.3 COUNTERFACTUAL EXPLANATIONS

The term "counterfactual" captures notions of what would have happened if something had been different. Two meanings have been used by ML subcommunities. 1) Those in causal inference make causal assumptions about interdependencies among variables and use these assumptions to incorporate consequential adjustments when particular variables are set to new values (Kusner et al., 2017; Pearl, 2019). 2) In contrast, the interpretability community recently used "counterfactual explanations" to explore how input variables must be modified to change a model's output *without* making explicit causal assumptions (Wachter et al., 2018). As such, counterfactual explanations can be seen as a case of contrastive explanations (Dhurandhar et al., 2018; Byrne, 2019). In this work, we use "counterfactual" in a sense similar to 2): we seek to make small changes to an input in order to reduce the uncertainty assigned to it by our model, without explicit causal assumptions.

Multiple counterfactual explanations can exist for any given input, as the functions we are interested in explaining are often non-injective (Russell, 2019). We are concerned with counterfactual input configurations that are close to the original input $\mathbf{x}_0$ according to some pairwise distance metric $d(\cdot, \cdot)$. Given a desired outcome $c$ different from the original one $\mathbf{y}_0$ produced by predictor $p_I$, counterfactual explanations $\mathbf{x}_c$ are usually generated by solving an optimization problem that resembles:

$$\mathbf{x}_c = \arg\max_{\mathbf{x}} \left( p_I(\mathbf{y}{=}c|\mathbf{x}) - d(\mathbf{x}, \mathbf{x}_0) \right) \quad \text{s.t.} \quad \mathbf{y}_0 {\neq} c. \tag{3}$$

Naively optimizing (3) in high-dimensional input spaces may result in the creation of adversarial inputs which are not actionable (Goodfellow et al., 2015). Telling a person that they would have been approved for a loan had their age been $-10$ is of very little use. To right this, recent works define linear constraints on explanations (Ustun et al., 2019; Sharma et al., 2020). An alternative more amenable to high dimensional data is to leverage DGMs (which we dub *auxiliary DGMs*) to ensure explanations are in-distribution (Dhurandhar et al., 2018; Joshi et al., 2018; Chang et al., 2019; Booth et al., 2020; Tripp et al., 2020). CLUE avoids the above issues by searching for counterfactuals in the lower-dimensional latent space of an auxiliary DGM. This choice is well suited for uncertainty, as the DGM constrains CLUE's search space to the data manifold. When faced with an OOD input, CLUE returns the nearest in-distribution analog, as shown in Figure 3.

## 3 PROPOSED METHOD

Without loss of generality, we use $\mathcal{H}$ to refer to any differentiable estimate of uncertainty ($\sigma$ or $H$). We introduce an auxiliary latent variable DGM: $p_\theta(\mathbf{x}) = \int p_\theta(\mathbf{x}|\mathbf{z})p(\mathbf{z})\,d\mathbf{z}$. In the rest of this paper, we will use the decoder from a variational autoencoder (VAE). Its encoder is denoted as $q_\phi(\mathbf{z}|\mathbf{x})$. We write these models' predictive means as $\mathbb{E}_{p_\theta(\mathbf{x}|\mathbf{z})}[\mathbf{x}]{=}\mu_\theta(\mathbf{x}|\mathbf{z})$ and $\mathbb{E}_{q_\phi(\mathbf{z}|\mathbf{x})}[\mathbf{z}]{=}\mu_\phi(\mathbf{z}|\mathbf{x})$ respectively.

CLUE aims to find points in latent space which generate inputs similar to an original observation $\mathbf{x}_0$ but are assigned low uncertainty. This is achieved by minimizing (4). CLUEs are then decoded as (5).

$$\mathcal{L}(\mathbf{z}) = \mathcal{H}(\mathbf{y}|\mu_\theta(\mathbf{x}|\mathbf{z})) + d(\mu_\theta(\mathbf{x}|\mathbf{z}), \mathbf{x}_0), \tag{4}$$

$$\mathbf{x}_{\text{CLUE}} = \mu_\theta(\mathbf{x}|\mathbf{z}_{\text{CLUE}}) \quad \text{where} \quad \mathbf{z}_{\text{CLUE}} = \arg\min_{\mathbf{z}}\mathcal{L}(\mathbf{z}). \tag{5}$$

The pairwise distance metric takes the form $d(\mathbf{x}, \mathbf{x}_0){=}\lambda_x d_x(\mathbf{x}, \mathbf{x}_0) + \lambda_y d_y(f(\mathbf{x}), f(\mathbf{x}_0))$ such that we can enforce similarity between uncertain points and CLUEs in both input and prediction space. The hyperparameters $(\lambda_x, \lambda_y)$ control the trade-off between producing low uncertainty CLUEs and CLUEs which are close to the original inputs. In this work, we take $d_x(\mathbf{x}, \mathbf{x}_0){=}\|\mathbf{x} - \mathbf{x}_0\|_1$ to encourage sparse explanations. For regression, $d_y(f(\mathbf{x}), f(\mathbf{x}_0))$ is mean squared error. For classification, we use cross-entropy. Note that the best choice for $d(\cdot, \cdot)$ will be task-specific.

The CLUE algorithm and a diagram of our procedure are provided in Algorithm 1 and Figure 4, respectively. The hyperparameter $\lambda_x$ is selected by cross validation for each dataset such that both terms in (4) are of similar magnitude. We set $\lambda_y$ to 0 for our main experiments, but explore different values in Appendix H.1. We minimize (4) with Adam by differentiating through both our BNN and VAE decoder. To facilitate optimization, the initial value of $\mathbf{z}$ is chosen to be $\mathbf{z}_0{=}\mu_\phi(\mathbf{z}|\mathbf{x}_0)$. Optimization runs for a minimum of three iterations and a maximum of 35 iterations, with a learning rate of 0.1. If the decrease in $\mathcal{L}(\mathbf{z})$ is smaller than $\mathcal{L}(\mathbf{z}_0)/100$ for three consecutive iterations, we apply early stopping. CLUE can be applied to batches of inputs simultaneously, allowing us to leverage GPU-accelerated matrix computation. Our implementation is detailed in full in Appendix B.

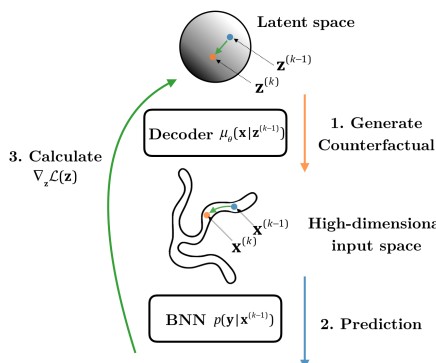

**Algorithm 1:** CLUE

**Inputs:** original datapoint $\mathbf{x}_0$, distance function $d(\cdot, \cdot)$, Uncertainty estimator $\mathcal{H}$, DGM decoder $\mu_\theta(\cdot)$, DGM encoder $\mu_\phi(\cdot)$

1 Set initial value of $\mathbf{z} = \mu_\phi(\mathbf{z}|\mathbf{x}_0)$;
2 **while** *loss $\mathcal{L}$ is not converged* **do**
3     Decode: $\mathbf{x} = \mu_\theta(\mathbf{x}|\mathbf{z})$;
4     Use predictor to obtain $\mathcal{H}(\mathbf{y}|\mathbf{x})$ ;
5     $\mathcal{L} = \mathcal{H}(\mathbf{y}|\mathbf{x}) + d(\mathbf{x}, \mathbf{x}_0)$;
6     Update $\mathbf{z}$ with $\nabla_{\mathbf{z}} \mathcal{L}$;
7 **end**
8 Decode explanation: $\mathbf{x}_{\text{CLUE}} = \mu_\theta(\mathbf{x}|\mathbf{z})$;
**Output:** Uncertainty counterfactual $\mathbf{x}_{\text{CLUE}}$

Figure 4: Latent codes are decoded into inputs for which a BNN generates uncertainty estimates; their gradients are backpropagated to latent space.

As noted by Wachter et al. (2018), individual counterfactuals may not shed light on all important features. Fortunately, we can exploit the non-convexity of CLUE's objective to address this. We initialize CLUE with $\mathbf{z}_0 = \mu_\phi(\mathbf{z}|\mathbf{x}_0) + \epsilon$, where $\epsilon = \mathcal{N}(\mathbf{z}; \mathbf{0}, \sigma_0 \mathbf{I})$, and perform Algorithm 1 multiple times to obtain different CLUEs. We find $\sigma_0 = 0.15$ to give a good trade-off between optimization speed and CLUE diversity. Appendix C shows examples of different CLUEs obtained for the same inputs.

We want to ensure noise from auxiliary DGM reconstruction does not affect CLUE visualization. For tabular data, we use the change in percentile of each input feature with respect to the training distribution as a measure of importance. We only highlight continuous variables for which CLUEs are separated by 15 percentile points or more from their original inputs. All changes to discrete variables are highlighted. For images, we report changes in pixel values by applying a sign-preserving quadratic function to the difference between CLUEs and original samples:

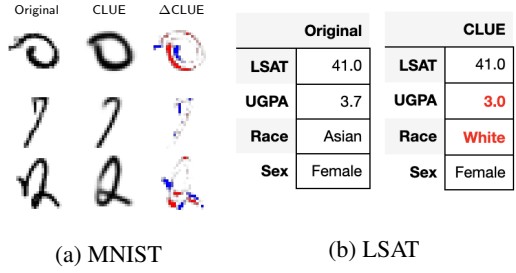

(a) MNIST        (b) LSAT

Figure 5: Example image and tabular CLUEs.

$\Delta\text{CLUE} = |\Delta\mathbf{x}| \cdot \Delta\mathbf{x}$ with $\Delta\mathbf{x} = \mathbf{x}_{\text{CLUE}} - \mathbf{x}_0$. This is showcased in Figure 5 and in Appendix G. It is common for approaches to generating saliency maps to employ constraints that encourage the contiguity of highlighted pixels (Chang et al., 2019; Dabkowski & Gal, 2017). We do not employ such constraints, but we note they might prove useful when applying CLUE to natural images.

## 4 A FRAMEWORK FOR EVALUATING COUNTERFACTUAL EXPLANATIONS OF UNCERTAINTY COMPUTATIONALLY

Evaluating explanations quantitatively (without resorting to expensive user studies) is a difficult but important task (Doshi-Velez & Kim, 2017; Weller, 2019). We put forth a computational framework to evaluate counterfactual explanations of uncertainty. In the spirit of Bhatt et al. (2020a), we desire counterfactuals that are *1) informative*: they should highlight features which affect our BNN's uncertainty, and *2) relevant*: counterfactuals should lie close to the original inputs and represent plausible parameter settings, lying close to the data manifold. Recall, from Figure 3, that inputs for which our BNN is certain can be constructed by applying adversarial perturbations to uncertain ones. Alas, evaluating these criteria requires access to the generative process of the data.

To evaluate the above requirements, we introduce an additional DGM that will act as a "ground truth" data generating process (g.t. DGM). Specifically, we use a variational autoencoder with arbitrary conditioning (Ivanov et al., 2019) (g.t. VAEAC). It jointly models inputs and targets $p_{\text{gt}}(\mathbf{x}, \mathbf{y})$. Measuring counterfactuals' log density under this model $\log p_{\text{gt}}(\mathbf{x}_c)$ allows us to evaluate if they are in-distribution. The g.t. VAEAC also allows us to query the conditional distribution over targets given inputs, $p_{\text{gt}}(\mathbf{y}|\mathbf{x})$. From this distribution, we can compute an input's true uncertainty $\mathcal{H}_{\text{gt}}$, as given by the generative process of the data. This allows us to evaluate if counterfactuals address the true

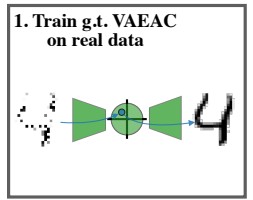 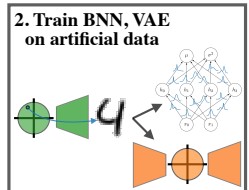 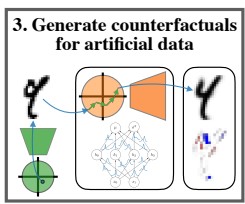 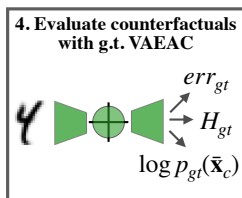

Figure 6: Pipeline for computational evaluation of counterfactual explanations of uncertainty. The VAEAC which we treat as a data generating process is colored in green. Colored in orange is the auxiliary DGM used by the approach being evaluated. For approaches that do not use an auxiliary DGM, like Uncertainty Sensitivity Analysis, the orange element will not be present.

sources of uncertainty in the data, as opposed to exploiting adversarial vulnerabilities in the BNN. The evaluation procedure, shown in Figure 6, is as follows:

1. Train a g.t. VAEAC on a real dataset to obtain $p_{\text{gt}}(\mathbf{x}, \mathbf{y})$ as well as conditionals $p_{\text{gt}}(\mathbf{y}|\mathbf{x})$.

2. Sample artificial data $(\bar{\mathbf{x}}, \bar{\mathbf{y}}) \sim p_{\text{gt}}(\mathbf{x}, \mathbf{y})$. Use them to train a BNN and an auxiliary DGM.

3. Sample more artificial data. Generate counterfactual explanations $\bar{\mathbf{x}}_c$ for uncertain samples.

4. Use the g.t. VAEAC to obtain the conditional distribution over targets given counterfactual inputs $p_{\text{gt}}(\mathbf{y}|\bar{\mathbf{x}}_c)$ and $\mathcal{H}_{\text{gt}}$. Evaluate if counterfactuals are on-manifold through $\log p_{\text{gt}}(\bar{\mathbf{x}}_c)$.

Given an uncertain artificially generated test point $\bar{\mathbf{x}}_0 \sim p_{\text{gt}}$ and its corresponding counterfactual explanation $\bar{\mathbf{x}}_c$, we quantify *informativeness* as the amount of uncertainty that has been explained away. The variance (or entropy) of $p_{\text{gt}}(\mathbf{y}|\mathbf{x})$ reflects the ground truth aleatoric uncertainty associated with $\mathbf{x}$. Hence, for aleatoric uncertainty, we quantify *informativeness* as $\Delta \mathcal{H}_{\text{gt}} = \mathbb{E}_{p_{\text{gt}}}[\mathcal{H}_{\text{gt}}(\mathbf{y}|\bar{\mathbf{x}}_0) - \mathcal{H}_{\text{gt}}(\mathbf{y}|\bar{\mathbf{x}}_c)]$. Epistemic uncertainty only depends on our BNN. It cannot be directly computed from $p_{\text{gt}}(\mathbf{y}|\mathbf{x})$. However, its reduction can be measured implicitly through the reduction in the BNN's prediction error with respect to the labels outputted by the g.t. VAEAC: $\Delta err_{\text{gt}} = \mathbb{E}_{p_{gt}}[err_{\text{gt}}(\bar{\mathbf{x}}_0) - err_{\text{gt}}(\bar{\mathbf{x}}_c)]$. Here $err_{\text{gt}}(\mathbf{x}) = d_y(p(y|\mathbf{x}), \arg\max_y p_{\text{gt}}(y|\mathbf{x}))$. Approaches that exploit adversarial weaknesses in the BNN will not transfer to the g.t. VAEAC, failing to reduce uncertainty or error. We assess the *relevance* of counterfactuals through their likelihood under the g.t. VAEAC $\log p_{gt}(\bar{\mathbf{x}}_c)$ and through their $\ell_1$ distance to the original inputs $\|\Delta \bar{\mathbf{x}}\|_1 = \|\bar{\mathbf{x}}_0 - \bar{\mathbf{x}}_c\|_1$. We refer to Appendix I for a detailed discussion on g.t. VAEACs and their use for comparing counterfactual generations.

## 5 EXPERIMENTS

We validate CLUE on LSAT academic performance regression (Wightman et al., 1998), UCI Wine quality regression, UCI Credit classification (Dua & Graff, 2017), a 7 feature variant of COMPAS recidivism classification (Angwin et al.), and MNIST image classification (LeCun & Cortes, 2010). For each, we select roughly the 20% most uncertain test points as those for which we reject our BNNs' decisions. We only generate CLUEs for "rejected" points. Rejection thresholds, architectures, and hyperparameters are in Appendix B. Experiments with non-Bayesian NNs are in Appendix H.1.

As a baseline, we introduce a localized version of Uncertainty Sensitivity Analysis (Depeweg et al., 2017). It produces counterfactuals by taking a single step, of size $\eta$, in the direction of the gradient of an input's uncertainty estimates $\mathbf{x}_c = \mathbf{x}_0 - \eta \nabla_{\mathbf{x}} \mathcal{H}(\mathbf{y}|\mathbf{x}_0)$. Averaging $|\mathbf{x}_0 - \mathbf{x}_c|$ across a test set, we recover (2). As a second baseline, we adapt FIDO (Chang et al., 2019), a counterfactual feature importance method, to minimize uncertainty. We dub this approach U-FIDO. This method places a binary mask $\mathbf{b}$ over the set of input variables $\mathbf{x}_U$. The mask is modeled by a product of Bernoulli random variables: $p_{\boldsymbol{\rho}}(\mathbf{b}) = \prod_{u \in U} \text{Bern}(b_u; \rho_u)$. The set of masked inputs $\mathbf{x}_B$ is substituted by its expectation under an auxiliary conditional generative model $p(\mathbf{x}_B|\mathbf{x}_{U \setminus B})$. We use a VAEAC. U-FIDO finds the masking parameters $\boldsymbol{\rho}$ which minimize (6):

$$\mathcal{L}(\boldsymbol{\rho}) = \mathbb{E}_{p_{\boldsymbol{\rho}}(\mathbf{b})}[\mathcal{H}(\mathbf{y}|\mathbf{x}_c(\mathbf{b})) + \lambda_b \|\mathbf{b}\|_1], \tag{6}$$

$$\mathbf{x}_c(\mathbf{b}) = \mathbf{b} \odot \mathbf{x}_0 + (1 - \mathbf{b}) \odot \mathbb{E}_{p(\mathbf{x}_B|\mathbf{x}_{U \setminus B})}[\mathbf{x}_B]. \tag{7}$$

Counterfactuals are generated by (7), where $\odot$ is the Hadamard product. We compare CLUE to feature importance methods (Ribeiro et al., 2016; Lundberg & Lee, 2017) in Appendix F.

## 5.1 COMPUTATIONAL EVALUATION

We compare CLUE, Localized Sensitivity, and U-FIDO using the evaluation framework put forth in Section 4. We would like counterfactuals to explain away as much uncertainty as possible while staying as close to the original inputs as possible. We manage this *informativeness* (large $\Delta \mathcal{H}_{\mathrm{gt}}$) to *relevance* (small $\|\Delta\bar{\mathbf{x}}\|_1$) trade-off with the hyperparameters $\eta$, $\lambda_x$, and $\lambda_b$ for Local Sensitivity, CLUE, and U-FIDO, respectively. We perform a logarithmic grid search over hyperparameters and plot Pareto-like curves. Our two metrics of interest take minimum values of $0$ but their maximum is dataset and method dependent. For Sensitivity, $\|\Delta\bar{\mathbf{x}}\|_1$ grows linearly with $\eta$. For CLUE and U-FIDO, these metrics saturate for large and small values of $\lambda_x$ (or $\lambda_b$). As a result, the values obtained by these methods do not overlap. As shown in Figure 7, CLUE is able to explain away more uncertainty ($\Delta \mathcal{H}_{\mathrm{gt}}$) than U-FIDO, and U-FIDO always obtains smaller values of $\|\Delta\bar{\mathbf{x}}\|_1$ than CLUE.

Table 1: $\Delta \mathcal{H}_{\mathrm{gt}}$ vs $\|\Delta\bar{\mathbf{x}}\|_1$ measure obtained by all methods on all datasets under consideration. Lower is better. The dimensionality of each dataset is listed next to their names. *e* and *a* indicate results for epistemic ($\Delta err_{\mathrm{gt}}$) and aleatoric ($\Delta \mathcal{H}_{\mathrm{gt}}$) uncertainty respectively.

Figure 7: MNIST knee-points.

| Method | LSAT (4) | | COMPAS (7) | | Wine (11) | | Credit (23) | | MNIST (784) | |
|---|---|---|---|---|---|---|---|---|---|---|
| | e | a | e | a | e | a | e | a | e | a |
| Sensitivity | 0.70 | 0.67 | **0.71** | **0.13** | 0.69 | 0.03 | 0.63 | 0.50 | 0.66 | 0.68 |
| CLUE | 0.52 | 0.64 | **0.71** | 0.18 | **0.01** | 0.14 | 0.52 | **0.29** | **0.26** | **0.27** |
| U-FIDO | **0.36** | **0.51** | **0.71** | 0.31 | 0.22 | **0.02** | **0.45** | 0.63 | 0.38 | 0.50 |

To construct a single performance metric, we scale all measurements by the maximum values obtained between U-FIDO or CLUE, e.g. $(\sqrt{2} \cdot \max(\Delta \mathcal{H}_{\mathrm{gt\ U\text{-}FIDO}}, \Delta \mathcal{H}_{\mathrm{gt\ CLUE}}))^{-1}$, linearly mapping them to $[0, 1/\sqrt{2}]$. We then negate $\Delta \mathcal{H}_{\mathrm{gt}}$, making its optimum value $0$. We consider each method's best performing hyperparameter configuration, as determined by its curve's point nearest the origin, or *knee-point*. The euclidean distance from each method's knee-point to the origin acts as a metric of relative performance. The best value is $0$ and the worst is $1$. Knee-point distances, computed across three runs, are shown for both uncertainty types in Table 1.

Local Sensitivity performs poorly on all datasets except COMPAS. We attribute this to the implicit low dimensionality of COMPAS: only two features are necessary to accurately predict targets (Dressel & Farid, 2018). U-FIDO's masking mechanism allows for counterfactuals that leave features unchanged. It performs well in low dimensional problems but suffers from variance as dimensionality grows. We conjecture that optimization in latent (instead of input) space makes CLUE robust to data complexity.

We perform an analogous experiment where *relevance* is quantified as proximity to the data manifold: $\Delta \log p_{gt} = \min(0, \log p_{gt}(\bar{\mathbf{x}}_c) - \log p_{gt}(\bar{\mathbf{x}}_0))$. Here, $\log p_{gt}(\bar{\mathbf{x}}_0)$ refers to the log-likelihood of the artificial data for which counterfactuals are generated. Results are in Table 2, where CLUE performs best in $8/10$ tasks. Generating counterfactuals with a VAE ensures that CLUEs are *relevant*. In Appendix H.2, we perform an analogous *informativeness* vs *relevance* experiment on real data. We obtain similar results to our computational evaluation framework, validating its reliability.

Table 2: $\Delta \log p_{gt}$ vs $\|\Delta\bar{\mathbf{x}}\|_1$ measure obtained by all methods on all datasets under consideration. Lower is better. *e* and *a* indicate epistemic ($\Delta err_{\mathrm{gt}}$) and aleatoric ($\Delta \mathcal{H}_{\mathrm{gt}}$) uncertainty respectively.

| Method | LSAT (4) | | COMPAS (7) | | Wine (11) | | Credit (23) | | MNIST (784) | |
|---|---|---|---|---|---|---|---|---|---|---|
| | e | a | e | a | e | a | e | a | e | a |
| Sensitivity | 0.697 | 0.672 | **0.707** | 0.122 | 0.691 | **0.001** | 0.623 | 0.454 | 0.682 | 0.698 |
| CLUE | 0.419 | **0.070** | **0.707** | **0.044** | **0** | 0.128 | **0** | **0.009** | **0.273** | **0.146** |
| U-FIDO | **0** | **0** | **0.707** | 0.303 | 0.224 | **0** | 0.233 | 0.628 | 0.450 | 0.516 |

## 5.2 USER STUDY

Human-based evaluation is a key step in validating the utility of tools for ML explainability (Hoffman et al., 2018). We want to assess the extent to which CLUEs help machine learning practitioners identify sources of uncertainty in ML models compared to using simple linear approximations (Local Sensitivity) or human intuition. To do this, we propose a forward-simulation task (Doshi-Velez & Kim, 2017), focusing on an appropriate local test to evaluate CLUEs. We show practitioners one

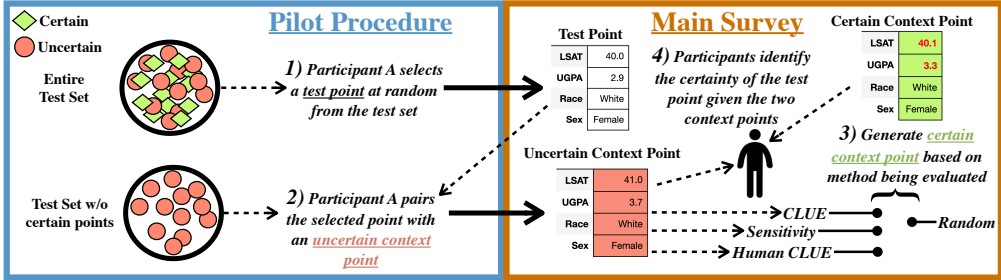

Figure 8: Experimental workflow for our tabular data user study.

| | Uncertain | | Certain | | ? |
|---|---|---|---|---|---|
| **Age** | Less than 25 | **Age** | Less than 25 | **Age** | Less than 25 |
| **Race** | Caucasian | **Race** | African-American | **Race** | Hispanic |
| **Sex** | Male | **Sex** | Male | **Sex** | Male |
| **Current Charge** | Misdemeanour | **Current Charge** | Misdemeanour | **Current Charge** | Misdemeanour |
| **Reoffended Before** | Yes | **Reoffended Before** | No | **Reoffended Before** | No |
| **Prior Convictions** | 1 | **Prior Convictions** | 0 | **Prior Convictions** | 0 |
| **Days Served** | 0 | **Days Served** | 0 | **Days Served** | 0 |

Figure 9: Example question shown to main survey participants for the COMPAS dataset: *Given the uncertain example on the left and the certain example in the middle, will the model be certain on the test example on the right?* The red text highlights the features that differ between *context points*.

datapoint below our "rejection" threshold and one datapoint above. The former is labeled as "certain" and the latter as "uncertain"; we refer to these as *context points*. The certain *context point* serves as a local counterfactual explanation for the uncertain *context point*. Using both *context points* for reference, practitioners are asked to predict whether a new test point will be above or below our threshold (i.e., will our BNN's uncertainty be high or low for the new point). Our survey compares the utility of the certain *context points* generated by CLUE relative to those from baselines.

In our survey, we compare four different methods, varying how we select certain *context points*. We either 1) select a certain point at random from the test set as a control, generate a counterfactual certain point with 2) Local Sensitivity or with 3) CLUE, or 4) display a human selected certain point (*Human CLUE*). To generate a *Human CLUE*, we ask participants (who will not take the main survey) to pair uncertain *context points* with similar certain points. We select the points used in our main survey with a pilot procedure similar to Grgic-Hlaca et al. (2018). This procedure, shown in Figure 8, prevents us from injecting biases into point selection and ensures *context points* are relevant to test points. In our procedure, a participant is shown a pool of randomly selected certain and uncertain points. We ask this participant to select points from this pool: these will be test points. We then ask the participant to map each selected test point to a similar uncertain point without replacement. In this way, we obtain uncertain *context points* that are relevant to test points.

We use the LSAT and COMPAS datasets in our user study. Ten different participants take each variant of the main survey: our participants are ML graduate students, who serve as proxies for practitioners in industry. The main survey consists of $18$ questions, $9$ per dataset. An example question is shown in Figure 9. The average participant accuracy by variant is: CLUE ($82.22\%$), *Human CLUE* ($62.22\%$), Random ($61.67\%$), and Local Sensitivity ($52.78\%$). We measure the statistical significance of CLUE's superiority with unpaired Wilcoxon signed-rank tests (Demšar, 2006) of CLUE vs each baseline. We obtain the following p-values: Human-CLUE ($2.34e-5$), Random ($1.47e-5$), and Sensitivity ($2.60e-9$).[1] Additional analysis is included in Appendix H.3.

---

[1] Our between-group experimental design could potentially result in dependent observations, which violates the tests' assumptions. However, the p-values obtained are extremely low, providing confidence in rejecting the null hypothesis of equal performance among approaches.

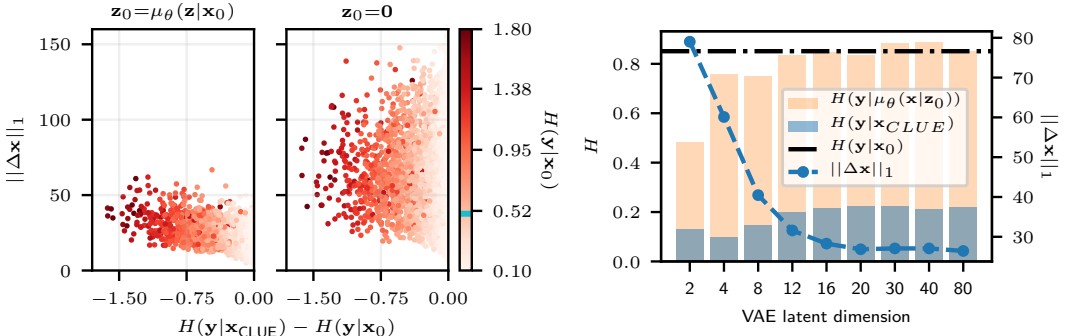

Figure 10: Left: CLUEs are similarly *informative* under encoder-based and encoder-free initializations. The colorbar indicates the original samples' uncertainty. Its horizontal blue line denotes our rejection threshold. Right: Auxiliary DGMs with more capacity result in more *relevant* CLUEs.

We find that linear explanations (Local Sensitivity) of a non-linear function (BNN) mislead practitioners and perform worse than random. While *Human CLUE* explanations are real datapoints, CLUE generates explanations from a VAE. We conjecture that CLUE's increased flexibility produces relevant explanations in a broader range of cases. In our tabular data user study, we only show one pair of *context points* per test point. We find that otherwise the survey is difficult for practitioners to follow, due to non-expertise in college admissions or criminal justice. Using MNIST, we run a smaller scale study, wherein we show participants larger sets of *context points*. Results are in Appendix J.2.

## 5.3 ANALYSIS OF CLUE'S AUXILIARY DEEP GENERATIVE MODEL

We study CLUE's reliance on its auxiliary DGM. Further ablative analysis is found in Appendix H.

**Initialization Strategy:** We compare Algorithm 1's encoder-based initialization $\mathbf{z}_0 = \mu_\phi(\mathbf{z}|\mathbf{x}_0)$ with $\mathbf{z}_0 = \mathbf{0}$. As shown in Figure 10, for high dimensional datasets, like MNIST, initializing $\mathbf{z}$ with the encoder's mean leads to CLUEs that require smaller changes in input space to explain away similar amounts of uncertainty (i.e., more *relevant*). In Appendix H.1, similar behavior is observed for Credit, our second highest dimensional dataset. On other datasets, both approaches yield indistinguishable results. CLUEs could plausibly be generated with differentiable DGMs that lack an encoding mechanism, such as GANs. However, an appropriate initialization strategy should be employed.

**Capacity of CLUE's DGM:** Figure 10 shows how auto-encoding uncertain MNIST samples with low-capacity VAEs significantly reduces these points' predictive entropy. CLUEs generated with these VAEs highlight features that the VAEs are unable to reproduce but are not reflective of our BNN's uncertainty. This results in large values of $\|\Delta\mathbf{x}\|_1$; although the counterfactual examples are indeed more certain than the original samples, they contain unnecessary changes. As our auxiliary DGMs' capacity increases, the amount of uncertainty preserved when auto-encoding inputs increases as well. $\|\Delta\mathbf{x}\|_1$ decreases while the predictive entropy of our CLUEs stays the same. More expressive DGMs allow for generating sparser, more *relevant*, CLUEs. Fortunately, even in scenarios where our predictor's training dataset is limited, we can train powerful DGMs by leveraging unlabeled data.

## 6 CONCLUSION

With the widespread adoption of data-driven decision making has come a need for the development of ML tools that can be trusted by their users. This has spawned a subfield of ML dedicated to interpreting deep learning systems' predictions. A transparent deep learning method should also inform stakeholders when it does not know the correct prediction (Bhatt et al., 2021). In turn, this creates a need for being able to interpret why deep learning methods are uncertain.

We address this issue with Counterfactual Latent Uncertainty Explanations (CLUE), a method that reveals which input features can be changed to reduce the uncertainty of a probabilistic model, like a BNN. We then turn to assessing the utility, to stakeholders, of counterfactual explanations of predictive uncertainty. We put forth a framework for computational evaluation of these types of explanations. Quantitatively, CLUE outperforms simple baselines. Finally, we perform a user study. It finds that users are better able to predict their models' behavior after being exposed to CLUEs.

ACKNOWLEDGEMENTS

JA acknowledges support from Microsoft Research through its PhD Scholarship Program. UB acknowledges support from DeepMind and the Leverhulme Trust via the Leverhulme Centre for the Future of Intelligence (CFI) and from the Mozilla Foundation. AW acknowledges support from a Turing AI Fellowship under grant EP/V025379/1, The Alan Turing Institute under EPSRC grant EP/N510129/1 & TU/B/000074, and the Leverhulme Trust via CFI.

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

## APPENDIX

This appendix is formatted as follows.

1. We discuss the **datasets** used in Appendix A.
2. **Implementation details** for our experiments are provided in Appendix B.
3. We provide examples of the **multiplicity of CLUEs** in Appendix C.
4. We discuss the application of **uncertainty sensitivity analysis** in high dimensional spaces in Appendix D.
5. We visualize CLUE's **optimization in the latent space** in Appendix E.
6. We compare CLUE to existing **feature importance** techniques in Appendix F.
7. We provide additional **examples** of CLUEs and U-FIDO counterfactuals in Appendix G.
8. We provide additional **experimental results** in Appendix H.
9. We note additional details of our **computational evaluation** framework for counterfactual explanations of uncertainty in Appendix I.
10. We include more details on the **setup** of our user studies in Appendix J.

# A    DATASETS

We employ 5 datasets in our experiments, 4 tabular and one composed of images. All of them are publicly available. Their details are given in Table 3.

Table 3: Summary of datasets used in our experiments. (*) We use a 7 feature version of COMPAS, however, other versions exist.

| Name | Targets | Input Type | N. Inputs | N. Train | N. Test |
|------|---------|------------|-----------|----------|---------|
| LSAT | Continuous | Continuous & Categorical | 4 | 17432 | 4358 |
| COMPAS | Binary | Continuous & Categorical | 7* | 5554 | 618 |
| Wine (red) | Continuous | Continuous | 11 | 1438 | 160 |
| Credit | Binary | Continuous & Categorical | 24 | 27000 | 3000 |
| MNIST | Categorical | Image (greyscale) | $28 \times 28$ | 60000 | 10000 |

We use the LSAT loading script from Cole & Williamson (2019)'s github page. The raw data can be downloaded from (`https://raw.githubusercontent.com/throwaway20190523/MonotonicFairness/master/data/law_school_cf_test.csv`) and (`https://raw.githubusercontent.com/throwaway20190523/MonotonicFairness/master/data/law_school_cf_train.csv`).

For the COMPAS criminal recidivism prediction dataset we use a modified version of Zafar et al. (2017)'s loading and pre-processing script. It can be found at (`https://github.com/mbilalzafar/fair-classification/blob/master/disparate_mistreatment/propublica_compas_data_demo/load_compas_data.py`).
We add an additional feature: "days served" which we compute as the difference, measured in days, between the "c_jail_in" and "c_jail_out" variables. The raw data is found at (`https://github.com/propublica/compas-analysis/blob/master/compas-scores-two-years.csv`).

The red wine quality prediction dataset can be obtained from and is described in detail at (`https://archive.ics.uci.edu/ml/datasets/wine+quality`).

The default of credit card clients dataset, which we refer to as "Credit" in this work, can be obtained from and is described in detail at (`https://archive.ics.uci.edu/ml/datasets/default+of+credit+card+clients`). Note that this dataset is different from the also commonly used German credit dataset.

The MNIST handwritten digit image dataset can be obtained from (`http://yann.lecun.com/exdb/mnist/`).

# B    IMPLEMENTATION DETAILS

## B.1    INFERENCE IN BNNs

We choose a Monte Carlo (MC) based inference approach for our BNNs due to these not being limited to localized approximations of the posterior. Specifically, we make use of scale adapted SG-HMC (Springenberg et al., 2016), an approach to stochastic gradient Hamiltonian Monte Carlo with automatic hyperparameter discovery. This technique estimates the mass matrix and the noise introduced by stochasticity in the gradients using exponentially decaying moving average filters during the chain's burn-in phase. We use a fixed step size of $\epsilon = 0.01$ and batch sizes of $512$. We set a diagonal 0 mean Gaussian prior $p(\mathbf{w}) = \mathcal{N}(\mathbf{w}; \mathbf{0}, \sigma_w^2 \cdot I)$ over each layer of weights. We place a per-layer conjugate Gamma hyperprior over $\sigma_w^2$ with parameters $\alpha = \beta = 10$. We periodically update $\sigma_w^2$ for each layer using Gibbs sampling.

On MNIST, we burn in our chain for 25 epochs, using the first 15 to estimate SG-HMC parameters. We re-sample momentum parameters every 10 steps and perform a Gibbs sweep over the prior variances every 45 steps. We save parameter settings every 2 epochs until a total of 300 sets of weights are stored. This makes for a total of 625 epochs.

For tabular datasets, we perform a burn-in of 400 epochs, using the first 120 to estimate SG-HMC parameters. We save weight configurations every 20 epochs until a total of 100 sets if weights are saved. This makes for a total of 2500 epochs. Momentum is re-sampled every 10 epochs and the prior over weights is re-sampled every 50 epochs. We use a batch size of 512 for all datasets.

## B.2    COMPUTING UNCERTAINTY ESTIMATES

In this work, we consider NNs which parametrize two types of distributions over target variables: the categorical for classification problems and the Gaussian for regression. For classification, our networks output a probability vector with elements $f_k(\mathbf{x}, \mathbf{w})$, corresponding to classes $\{c_k\}_{k=1}^K$. The likelihood function is $p(y|\mathbf{x}, \mathbf{w}) = Cat(y; f(\mathbf{x}, \mathbf{w}))$. Given a posterior distribution over weights $p(\mathbf{w}|\mathcal{D})$, we use marginalization (1) to translate uncertainty in $\mathbf{w}$ into uncertainty in predictions. Unfortunately, this operation is intractable for BNNs. We resort to approximating the predictive posterior with $M$ MC samples:

$$p(\mathbf{y}^*|\mathbf{x}^*, \mathcal{D}) = \mathbb{E}_{p(\mathbf{w}|\mathcal{D})}[p(\mathbf{y}^*|\mathbf{x}^*, \mathbf{w})]$$

$$\approx \frac{1}{M} \sum_{m=0}^{M} f(\mathbf{x}^*, \mathbf{w}); \quad \mathbf{w} \sim p(\mathbf{w}|\mathcal{D}).$$

The resulting predictive distribution is categorical. We quantify its uncertainty using entropy:

$$H(\mathbf{y}^*|\mathbf{x}^*, \mathcal{D}) = \sum_{k=1}^{K} p(y^*{=}c_k|\mathbf{x}^*, \mathcal{D}) \log p(y^*{=}c_k|\mathbf{x}^*, \mathcal{D}).$$

This quantity contains aleatoric and epistemic components $(H_a, H_e)$. The former is estimated as:

$$H_a = \mathbb{E}_{p(\mathbf{w}|\mathcal{D})}[H(y^*|\mathbf{x}^*, \mathbf{w})] \approx \frac{1}{M} \sum_{m}^{M} H(y^*|\mathbf{x}^*, \mathbf{w}); \quad \mathbf{w} \sim p(\mathbf{w}|\mathcal{D}).$$

The epistemic component can be obtained as the difference between the total and aleatoric entropies. This quantity is also known as the mutual information between $\mathbf{y}^*$ and $\mathbf{w}$:

$$H_e = I(\mathbf{y}^*, \mathbf{w}|\mathbf{x}^*, \mathcal{D}) = H(y^*|\mathbf{x}^*, \mathcal{D}) - \mathbb{E}_{p(\mathbf{w}|\mathcal{D})}[H(y^*|\mathbf{x}^*, \mathbf{w})].$$

For regression, we employ heteroscedastic likelihood functions. Their mean and variance are parametrized by our NN: $p(\mathbf{y}^*|\mathbf{x}^*, \mathbf{w}) = \mathcal{N}(\mathbf{y}; f_\mu(\mathbf{x}^*, \mathbf{w}), f_{\sigma^2}(\mathbf{x}^*, \mathbf{w}))$. Marginalizing over $\mathbf{w}$ with MC induces a Gaussian mixture distribution over outputs. Its mean is obtained as:

$$\boldsymbol{\mu}_a \approx \frac{1}{M} \sum_{m=0}^{M} f_\mu(\mathbf{x}^*, \mathbf{w}); \quad \mathbf{w} \sim p(\mathbf{w}|\mathcal{D}).$$

There is no closed-form expression for the entropy of this distribution. Instead, we use the variance of the GMM as an uncertainty metric. It also decomposes into aleatoric and epistemic components $(\sigma_a^2, \sigma_e^2)$:

$$\sigma^2(\mathbf{y}^*|\mathbf{x}^*, \mathcal{D}) = \underbrace{\mathbb{E}_{p(\mathbf{w}|\mathcal{D})}[\sigma^2(\mathbf{y}^*|\mathbf{x}^*, \mathbf{w})]}_{\sigma_a^2} + \underbrace{\sigma_{p(\mathbf{w}|\mathcal{D})}^2[\mu(\mathbf{y}^*|\mathbf{x}, \mathbf{w})]}_{\sigma_e^2}.$$

These are also estimated with MC:

$$\sigma^2(\mathbf{y}^*|\mathbf{x}^*, \mathcal{D}) \approx \underbrace{\frac{1}{M}\sum_m^M \mu(\mathbf{y}^*|\mathbf{x}^*, \mathbf{w})^2 - (\frac{1}{M}\sum_m^M \mu(\mathbf{y}^*|\mathbf{x}^*, \mathbf{w}))^2}_{\sigma_e^2} + \underbrace{\frac{1}{M}\sum_m^M \sigma^2(\mathbf{y}^*|\mathbf{x}^*, \mathbf{w})}_{\sigma_a^2}; \ \mathbf{w} \sim p(\mathbf{w}|\mathcal{D}).$$

Here, $\sigma_e^2$ reflects model uncertainty - our lack of knowledge about $\mathbf{w}$ - while $\sigma_a^2$ tells us about the irreducible uncertainty or noise in our training data.

In Figure 11, we show the fit obtained with a BNN with scale adapted SG-HMC on the toy moons dataset. We would like to highlight 2 key differences with respect to the logistic regression example shown in Figure 2. Neural networks are very flexible models. They are capable of perfectly fitting non-linear manifolds, such as moons. In consequence, when these models present *aleatoric uncertainty* it is most often due to the inputs not containing enough information to predict the targets. As little such noise exists in our particular instantiation of moons, our estimates of aleatoric entropy are close to 0. Despite their flexibility, selecting a NN involves adopting some inductive biases (Wilson & Izmailov, 2020). Additionally, unlike logistic regression, the weight space posterior of a BNN is very difficult to characterize. Both of these things are reflected in the BNN predictive posterior's *epistemic uncertainty* only growing in the vertical axis, instead of in all directions.

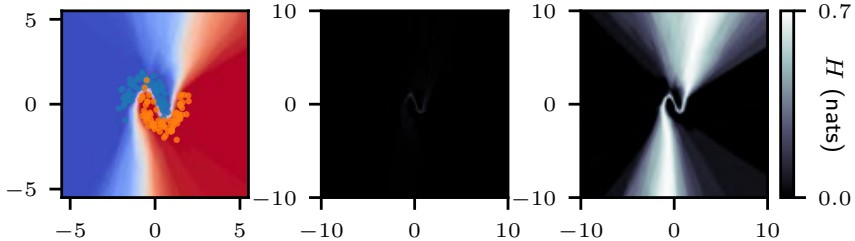

Figure 11: Left: Training points and BNN predictive distribution obtained on the moons dataset with SG-HMC. Center: Aleatoric entropy $H_a$ expressed by the model matches regions of class overlap. Right: Epistemic entropy $H_e$ grows as we move away from the data.

### B.3 ARCHITECTURES AND OTHER NETWORK HYPERPARAMETERS

For all datasets, our BNNs are fully connected networks with residual connections. Auxiliary VAEs and VAEACs used for tabular data use fully connected encoders and decoders with residual connections and batch normalization at every layer. For MNIST, we employ 6 convolutional bottleneck residual blocks (He et al., 2016) for both encoders and decoders. We use the same architecture for the VAEACs used as ground truth generative models in the computationally grounded evaluation framework put forth in Section 4. Note that the ground truth VAEAC models have slightly larger input spaces due to them modeling inputs and targets jointly. All architectural hyperparameters are provided in Table 4.

In order to improve the artificial sample quality of our "ground truth" VAEACs, we leverage a two-stage VAE configuration (Dai & Wipf, 2019). For all datasets, the lower level VAEs use the standard tabular data VAE architecture described above, with 2 hidden layers. We use 300 hidden units for MNIST and 150 for other datasets. Additional details on our use of two-stage VAEs are provided in Appendix I.

We train all generative models with the RAdam optimizer (Liu et al., 2020) with a learning rate of $1e^{-4}$ for tabular data and $3e^{-4}$ for MNIST. We found RAdam to yield marginally better results than Adam.

Table 4: Network architecture hyperparameters used in all experiments. Depth refers to number of hidden layers or residual blocks. Latent dimension values marked with a star (*) refer to the second level VAEs for "ground truth" VAEACs.

| Dataset | BNN Depth | BNN Width | VAE / VAEAC Depth | VAE Width | VAEAC Width | VAE / VAEAC Latent Dim |
|---------|-----------|-----------|-------------------|-----------|-------------|------------------------|
| LSAT | 2 | 200 | 3 | 300 | 350 | 4 (*4) |
| COMPAS | 2 | 200 | 3 | 300 | 350 | 4 (*4) |
| Wine | 2 | 200 | 3 | 300 | 350 | 6 (*6) |
| Credit | 2 | 200 | 3 | 300 | 350 | 8 (*8) |
| MNIST | 2 | 1200 | 6 | - | - | 20 (*8) |

We convert categorical inputs to our BNNs into one-hot vectors. When building DGMs, we model continuous inputs with diagonal, unit variance (heteroscedastic) Gaussian distributions. This choice makes these models weigh all input dimensions equally, a desirable trait for explanation generation. We place categorical distributions over discrete inputs, expressing them as one-hot vectors. For the LSAT, COMPAS, and Credit datasets, where there are both continuous and discrete features, data likelihood values are obtained as the product of Gaussian likelihoods and categorical likelihoods. During the CLUE optimization procedure, we approximate gradients through one-hot vectors with the softmax function's gradients. This is known as the softmax straight-through estimator (Bengio, 2013). It is biased but works well in practice. For MNIST, we model pixels as the probabilities of a product of Bernoulli distributions. We feed these probabilities directly into our BNNs and DGMs.

We normalize all continuously distributed features such that they have 0 mean and unit variance. This facilitates model training and also ensures that all features are weighed equally under CLUE's pairwise distance metric in (4). For MNIST, this normalization is applied to whole images instead of individual pixels. Categorical variables are not normalized. Changing a categorical variable implies changing two bits in the corresponding one-hot vector. This creates the same $\ell_1$ regularization penalty as shifting a continuously distributed variable two standard deviations.

## B.4 CLUE HYPERPARAMETERS

As mentioned in Section 5, in our experiments we only apply CLUE to points that present uncertainty above a rejection threshold. The rejection thresholds used for each dataset are displayed in Table 5. The same table contains the values of $\lambda_x$ used in all experiments. In practice we define $\lambda_x' = \lambda_x \cdot d$, where $d$ is the input space dimensionality of a dataset. This makes the strength of CLUE's pairwise input space distance metric agnostic to dimensionality. We choose a significantly larger value of $\lambda_x'$ for MNIST due to there being a large number of pixels that are always black.

Table 5: Values of CLUE's input space similarity weight $\lambda_x$ and uncertainty rejection thresholds used for all experiments. Next to each dataset's name is the the type of uncertainty quantified: standard deviation ($\sigma$) or entropy ($H$). We report $\lambda_x$ upscaled by each dataset's input dimensionality $d$.

| Dataset | LSAT ($\sigma$) | COMPAS ($H$) | Wine ($\sigma$) | Credit ($H$) | MNIST ($H$) |
|---------|-----------------|--------------|-----------------|--------------|-------------|
| $\lambda_x \cdot d$ | 1.5 | 2 | 2.5 | 3 | 25 |
| $\mathcal{H}$ threshold | 1 | 0.2 | 2 | 0.5 | 0.5 |

## C MULTIPLICITY OF CLUEs

We exploit the non-convexity of CLUE's objective to generate diverse CLUEs. We initialize CLUE with $\mathbf{z}_0 = \mu_\phi(\mathbf{z}|\mathbf{x}_0) + \epsilon$, where $\epsilon = \mathcal{N}(\mathbf{z}; \mathbf{0}, \sigma_0\mathbf{I})$, and perform Algorithm 1 multiple times to obtain different CLUEs. We choose $\sigma_0 = 0.15$. In Figure 12, we showcase different CLUEs for the same original MNIST inputs. Different counterfactuals represent digits of different classes. Despite this, all explanations resemble the original datapoints being explained. Being exposed to this multiplicity could potentially inform practitioners about similarities of an original input to multiple classes that lead their model to be uncertain.

Different initializations lead to CLUEs that explain away different amounts of uncertainty. In a few rare cases CLUE fails: the algorithm does not produce a feature configuration which has significantly lower uncertainty than the original input. This is the case for the third CLUE in the bottom 2 rows of Figure 12. We attribute this to a disadvantageous initialization of **z**.

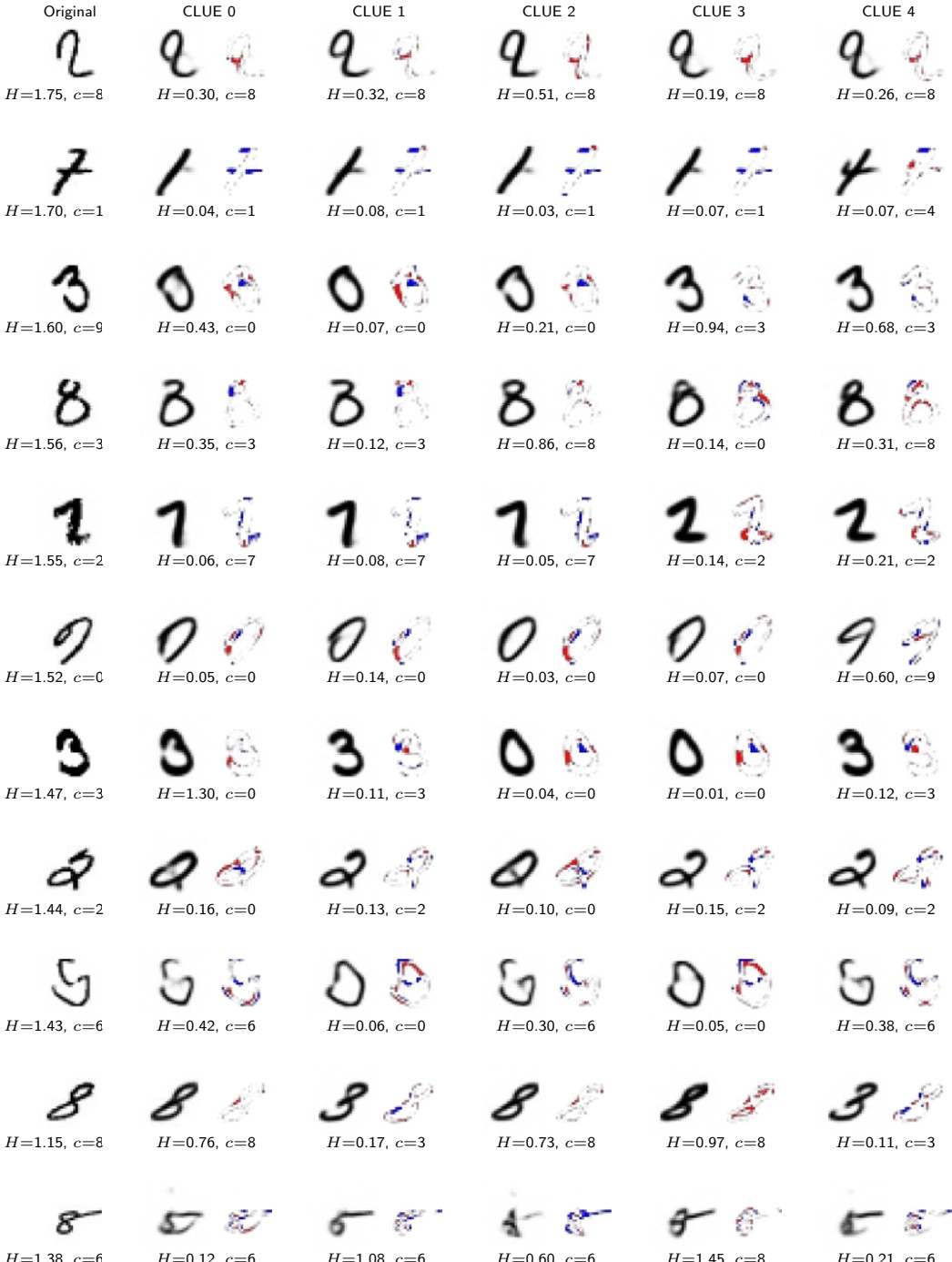

Figure 12: We generate 5 possible CLUEs for 11 MNIST digits score above the uncertainty rejection threshold. Below each digit or counterfactual is the predictive entropy it is assigned $H$ and the class of maximum probability $c$.

In Figure 13, we show multiple CLUEs for a single individual from the COMPAS dataset. In this case, uncertainty can be reduced by changing the individual's prior convictions and charge degree, or by changing their sex and age range. Making both sets of changes simultaneously also reduces uncertainty.

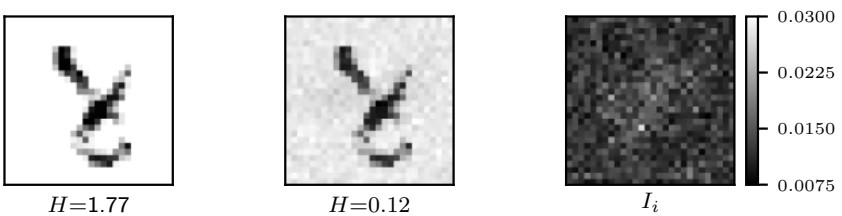

| | Original | | CLUE | | CLUE | | CLUE | | CLUE |
|---|---|---|---|---|---|---|---|---|---|
| **Age** | Greater than 45 | **Age** | Greater than 45 | **Age** | 25-45 | **Age** | 25-45 | **Age** | 25-45 |
| **Race** | African-American | **Race** | African-American | **Race** | African-American | **Race** | African-American | **Race** | Asian |
| **Sex** | Female | **Sex** | Female | **Sex** | Male | **Sex** | Male | **Sex** | Female |
| **Current Charge** | Felony | **Current Charge** | Misdemeanour | **Current Charge** | Misdemeanour | **Current Charge** | Felony | **Current Charge** | Felony |
| **Reoffended Before** | No | **Reoffended Before** | No | **Reoffended Before** | No | **Reoffended Before** | No | **Reoffended Before** | No |
| **Prior Convictions** | 1 | **Prior Convictions** | 0 | **Prior Convictions** | 0 | **Prior Convictions** | 1 | **Prior Convictions** | 1 |
| **Days Served** | 0 | **Days Served** | 0 | **Days Served** | 0 | **Days Served** | 0 | **Days Served** | 0 |

Figure 13: The leftmost entry is an uncertain COMPAS test sample. To its right are four candidate CLUEs. The first three successfully reduce uncertainty past our rejection threshold, while the rightmost does not.

## D  SENSITIVITY ANALYSIS IN HIGH DIMENSIONAL SPACES

In high-dimensional input spaces, $\nabla_{\mathbf{x}}\mathcal{H}$ will often not point in the direction of the data manifold. This can result in meaningless explanations. In Figure 14, we show an example where a step in the direction of $-\nabla_{\mathbf{x}}\mathcal{H}$ leads to a seemingly noisy input configuration for which the predictive entropy is low. An "adversarial examples for uncertainty" is generated. Aggregating these steps for every point in the test set leads to an uncertainty sensitivity analysis explanation that resembles white noise.

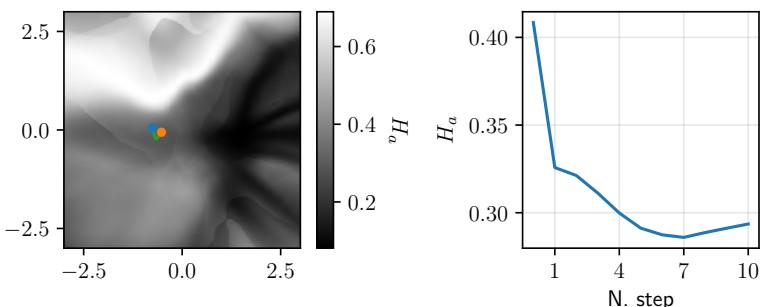

$H{=}1.77$                     $H{=}0.12$                     $I_i$

Figure 14: Left: A digit from the MNIST test set with large predictive entropy. Center: The same digit after a step is taken in the direction of $-\nabla_{\mathbf{x}}\mathcal{H}$. Non-zero weight is assigned to pixels that are always zero valued. Right: Uncertainty sensitivity analysis for the entire MNIST test set.

## E  VISUALIZING OPTIMIZATION IN LATENT SPACE

Figure 15 shows a 2 dimensional latent space trajectory from $\mathbf{z}_0$ to $\mathbf{z}_{CLUE}$ for a test point from the COMPAS dataset. In practice, we use larger latent spaces to ensure CLUEs are relevant.

Figure 15: Left: CLUE latent trajectory for a test point from the Credit dataset in a two-dimensional latent space. The blue dot marks the start of the trajectory and the orange one marks the end. Uncertainty levels are displayed in greyscale. Right: Changes in aleatoric entropy for inputs regenerated from latent codes along the trajectory.

# F    COMPARING CLUE TO FEATURE IMPORTANCE ESTIMATORS

Among machine learning practitioners, two of the most popular approaches for determining feature importance from back-box models are LIME and SHAP Bhatt et al. (2020b). LIME locally approx-imates the back-box model of interest around a specific test point with a surrogate linear model (Ribeiro et al., 2016). This surrogate is trained on points sampled from nearby the input of interest. The surrogate model's weights for each class can be interpreted as each feature's contribution towards the prediction of said class. Kernel SHAP extends lime by introducing a kernel such that resulting explanations have desirable properties (Lundberg & Lee, 2017). For SHAP, a reference input is chosen. It allows importance to be only assigned where the inputs are different from the reference. For MNIST, the reference is an entirely black image. Note that alternative versions of SHAP exist that incorporate information about internal NN dynamics into their explanations. However, they produce very noisy explanations when applied to our BNNs. We conjecture that this high variance might be induced by disagreement among the multiple weight configurations from our BNNs.

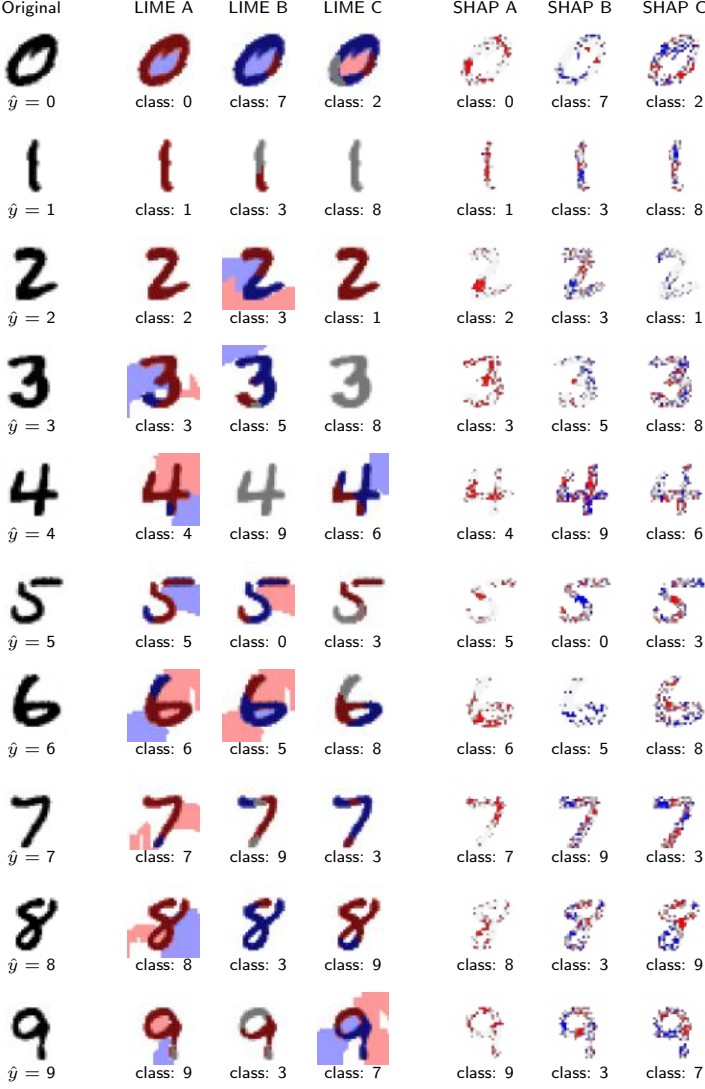

Figure 16: High confidence MNIST test examples together with LIME and SHAP explanations for the top 3 predicted classes. The model being investigated is a BNN with architecture described in Appendix B. The highest probability class is denoted by $\hat{y}$.

Figure 16 shows examples of LIME and Kernel SHAP being applied to a BNN for high confidence MNIST test digits. We use the default LIME hyperparameters for MNIST: the "quickshift" segmentation algorithm with kernel size 1, maximum distance 5 and a ratio of 0.2. We plot the top 10 segments with weight greater than 0.01. We draw 1000 samples with both methods.

Using the same configuration, we generate LIME and SHAP explanations for some MNIST digits to which our BNN assigns predictive entropy above our rejection threshold. The results are displayed in Figure 17.

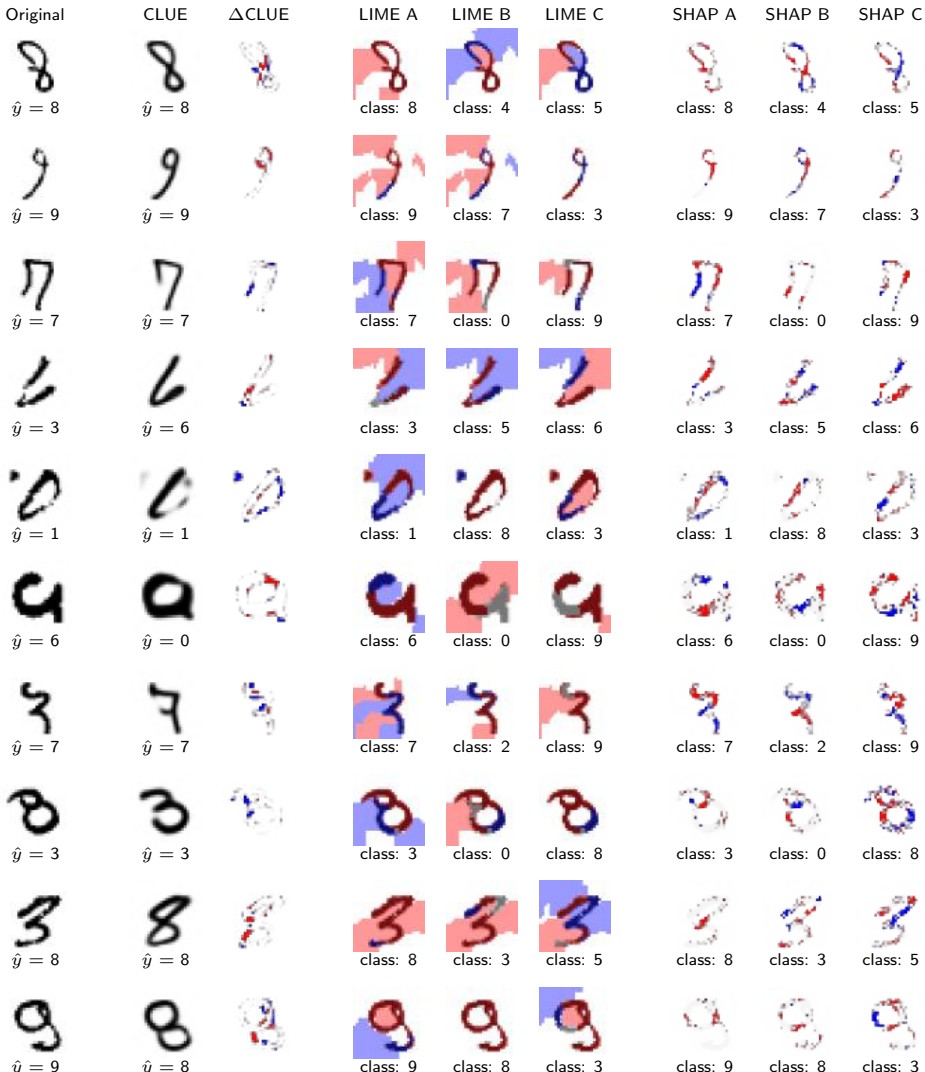

Figure 17: Ten MNIST test digits for which our BNN's predictive entropy is above the rejection threshold. A single CLUE example is provided for each one. For each digit, the top scoring class is denoted by $\hat{y}$. LIME and SHAP explanations are provided for the three most likely classes.

A positive CLUE attribution means that the addition of that feature will make our model more certain. A positive feature importance attribution means the presence of that feature serves as evidence towards a predicted class. A negative CLUE attribution means that the the absence of that feature will make the model more certain. A negative feature importance attribution means the absence of that feature would serve as evidence for a particular prediction. While CLUE and feature importance techniques solve similar problems and both provide saliency maps, CLUE highlights regions that need to be added or removed to make the input certain to a predictive model. In some cases, we see that feature importance negative attribution aligns with CLUE negative attribution, suggesting the

features which negatively contribute to the model's predicted probability are the features that need to be removed to increase the models' certainty. CLUE's ability to suggest the addition of unobserved features (positive CLUE attribution) is unique.

The feature importance methods under consideration are difficult to retrofit for uncertainty. They are unable to add features; they are limited to explaining the contribution of existing features. This may suffice if our input contains all the information needed to make a prediction for a certain class but otherwise results in noisy, potentially meaningless, explanations.

Generative-model based methods methods are counterfactual because they do not assign importance to the observed features but rather propose alternative features based on the data manifold Chang et al. (2019). This is the case for FIDO and CLUE. Generative modeling allows for increased flexibility, which is required when dealing with uncertain inputs. Quantitatively contrasting feature importance and uncertainty explanations under existing evaluation criteria Bhatt et al. (2020a) is an interesting direction for future work.

Methods like LIME and SHAP require a choice of class to produce explanations. This complicates their use in scenarios where our model is uncertain and multiple classes have similarly high predictive probability. On the other hand CLUEs are class agnostic.

## G    ADDITIONAL CLUE AND U-FIDO EXAMPLES

We provide additional examples of CLUEs generated for high uncertainty MNIST digits in Figure 18. U-FIDO counterfactuals generated for the same inputs are shown in Figure 19. Both methods often attribute importance to the same features. However, in almost all cases, CLUE is able to reduce the original input's uncertainty significantly more than U-FIDO. The latter method suggests smaller changes. We attribute this to U-FIDO's input masking mechanism being less flexible than CLUE's latent space generation mechanism.

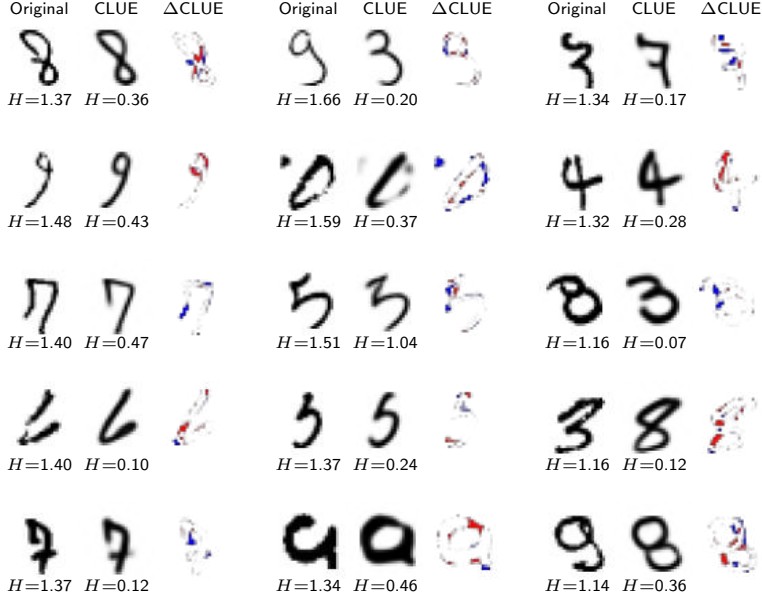

Figure 18: CLUEs generated for MNIST digits for which our BNN's predictive entropy is above the rejection threshold. The BNNs predictive entropy for both original inputs and CLUEs is shown under the corresponding images.

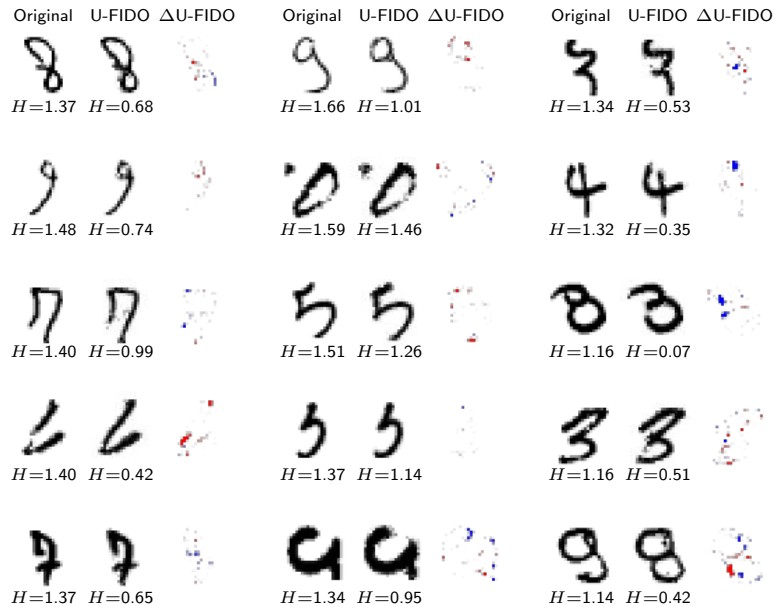

Figure 19: U-FIDO counterfactuals generated for MNIST digits for which our BNN's predictive entropy is above the rejection threshold. The BNNs predictive entropy for both original inputs and counterfactuals is shown under the corresponding images.

# H ADDITIONAL EXPERIMENTAL RESULTS

## H.1 ABLATION EXPERIMENTS

In this subsection, we modify some of CLUE's components individually and observe the effects on the procedure's results.

**Initialization Strategy:** Figure 20 compares Algorithm 1's encoder-based initialization $\mathbf{z}_0 = \mu_\phi(\mathbf{z}|\mathbf{x}_0)$ with $\mathbf{z}_0 = \mathbf{0}$ on all datasets under consideration. For the LSAT, COMPAS and Wine datasets, both approaches produce indistinguishable results. On Credit, our second highest dimensional dataset, using an encoder-based initialization allows for CLUEs to stay slightly closer to original inputs in terms of $\ell_1$ distance.

The difference between both approaches is largest on MNIST. We conjecture that this might be due to the higher dimensional nature of the latent space used with this dataset making optimization more difficult. By initializing $\mathbf{z}$ as the VAE encoder's mean, our optimizer starts near a local minima of $d(\mathbf{x}, \mathbf{x}_0)$ and potentially of $\mathcal{L}(\mathbf{z})$. When Algorithm 1 is applied, the magnitude of $\nabla_z H$ might not be large enough to escape this basin of attraction. Thus, CLUE tends to leave most input features unchanged, only addressing those with most potential to reduce uncertainty. This is also desirable behavior for low uncertainty inputs; the closest low uncertainty sample is the input itself.

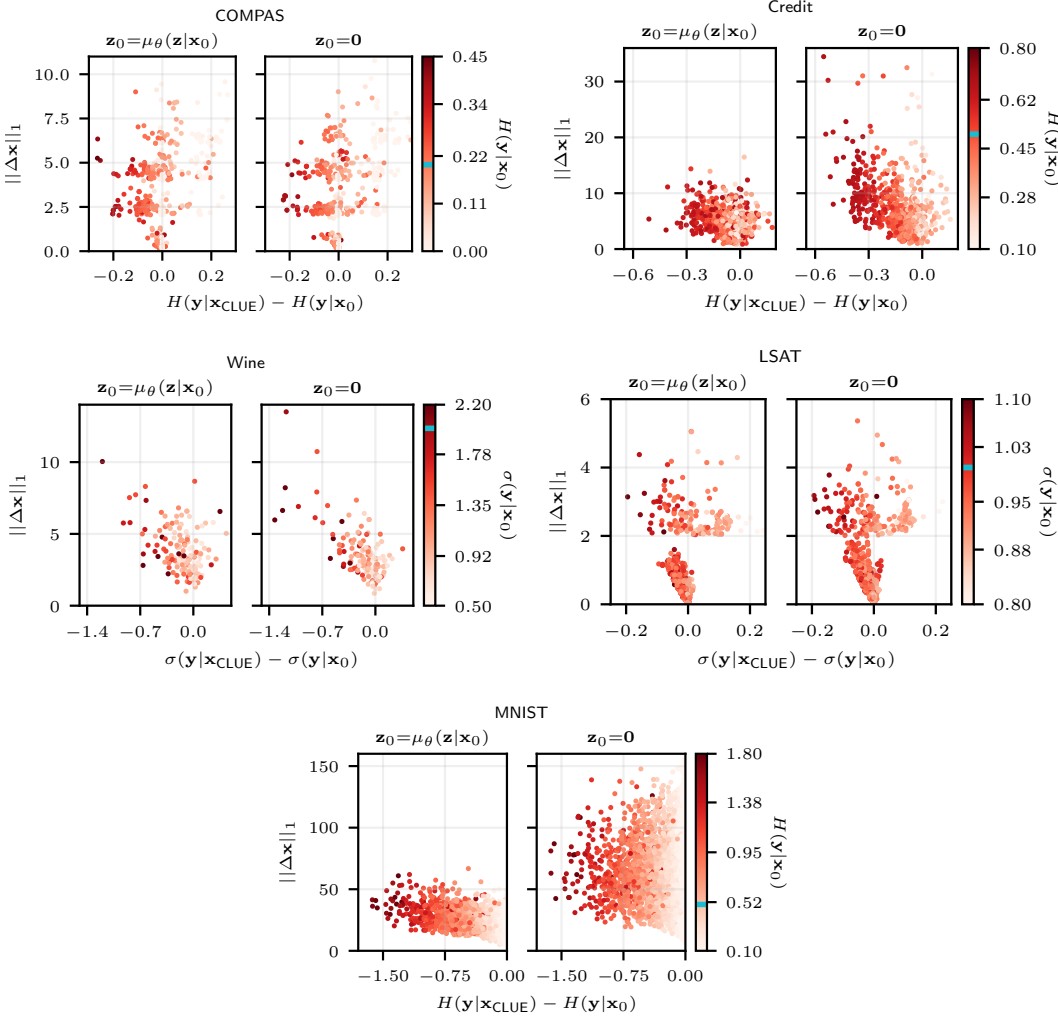

Figure 20: Initialization strategy experiment results for all datasets under consideration. Colorbars' horizontal blue line denotes each dataset's rejection threshold.

**Capacity of CLUE's DGM:** To capture our predictive model's reasoning, CLUE's DGM must be flexible enough to preserve atypical features in the inputs. As shown in Figure 21, reconstructions from low-capacity VAEs do not preserve the predictive uncertainty of original inputs. The CLUEs generated from these DGMs either leave the inputs unchanged or present large values of $\Delta\mathbf{x}$ while barely reducing $H$: these degenerate CLUEs simply emphasize regions of large reconstruction error. As our DGM's capacity increases, so does the amount of uncertainty preserved in the auto-encoding operation. The amount of predictive uncertainty explained by CLUEs, which is given by the difference between the autoencoded input uncertainty (orange bars) and CLUE uncertainty (blue bars), increases. We see a clear relationship between dataset dimensionality and size of latent space needed for CLUE to be effective.

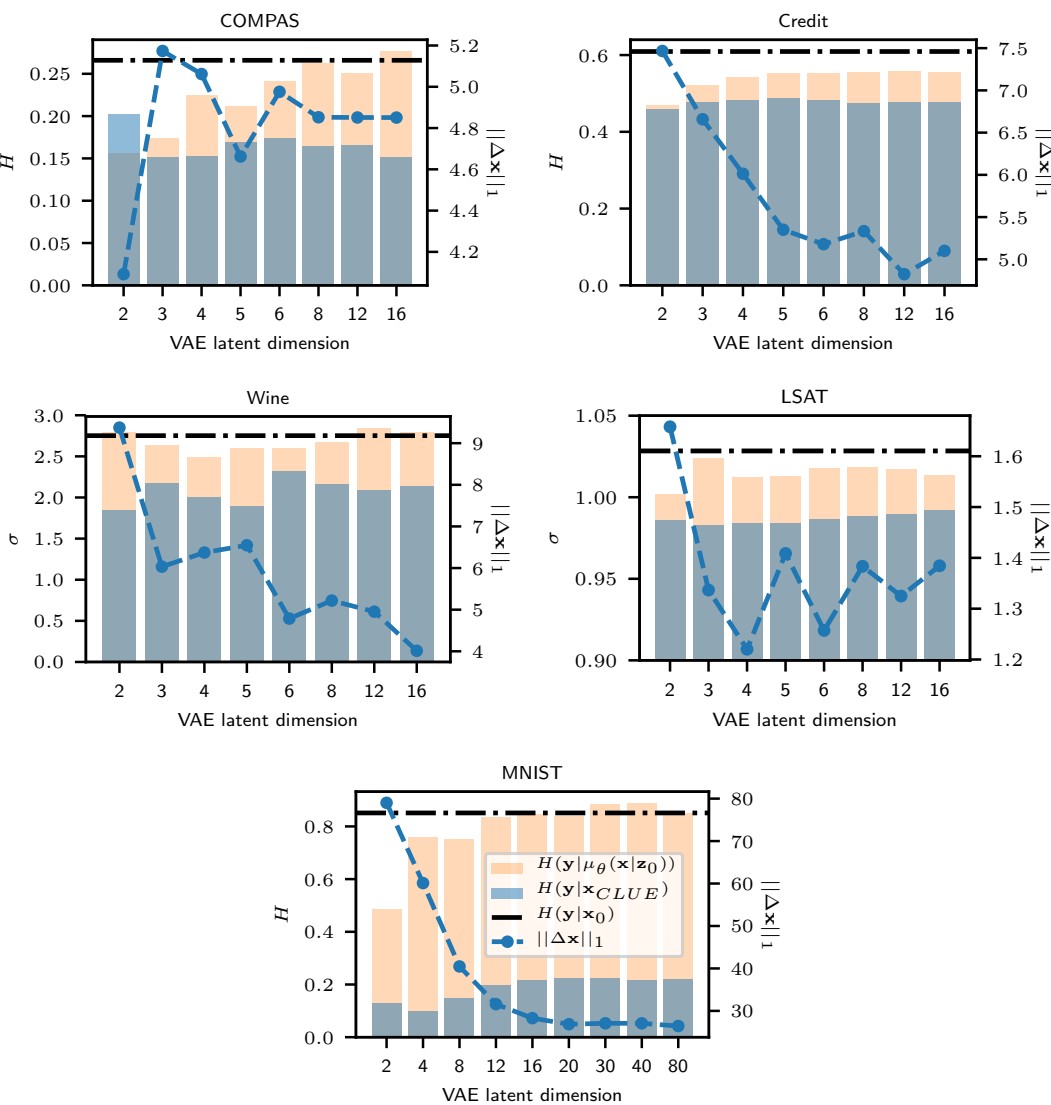

Figure 21: Amount of uncertainty explained away and $\ell_1$ distance between original inputs and CLUEs for every dataset under consideration and different capacity VAEs.

**Output Space Regularization Parameter** $\lambda_{\mathbf{y}}$: In Figure 22, we show how increasing $\lambda_y$ reduces the proportion of samples for which the predicted class differs between original inputs and CLUEs. Interestingly, on LSAT, Wine and COMPAS, a small, but non-zero, value of $\lambda_y$ results in more uncertainty being explained away by CLUE. However, strongly enforcing similarity of predictions generally comes at the cost of smaller amounts of uncertainty being explained away.

COMPAS predictions stay the same for all values of $\lambda_y$. Class predictions only depend on 2 of this dataset's input features (Age and Previous Convictions) (Dressel & Farid, 2018). We find that the remaining features can increase or reduce confidence in the prediction given by the two key features, but never change it. CLUEs only change non key features, reinforcing the current classification. On MNIST, we find that, for certain values of $\lambda_y$, classifying CLUEs results in a lower error rate than classifying original inputs. This is shown in Figure 23. We did not observe this effect for other datasets.

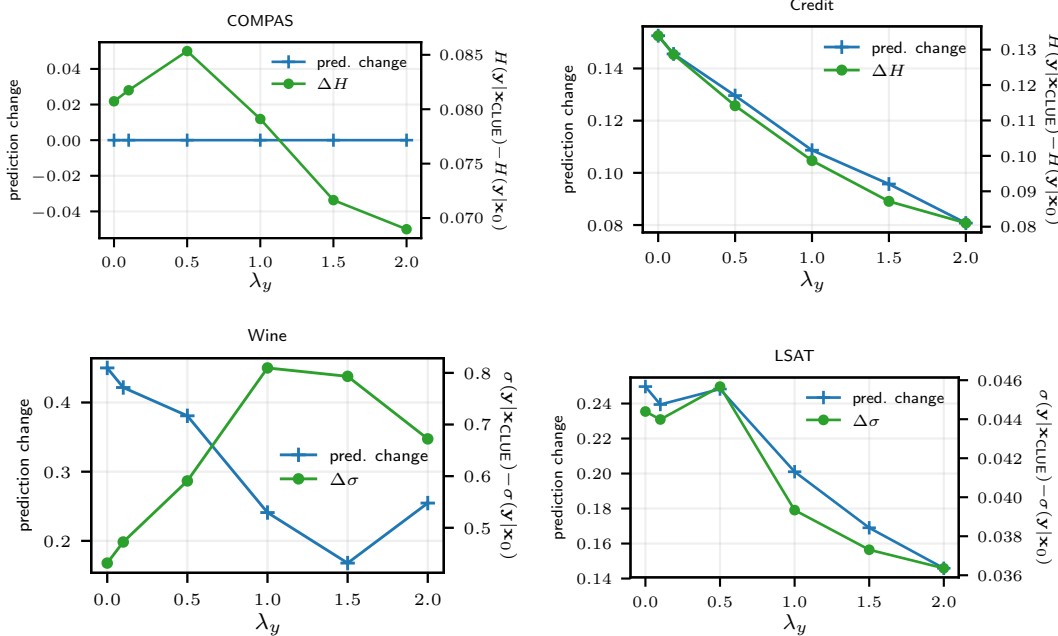

Figure 22: CLUE $\Delta\mathcal{H}$ vs prediction change for all datasets under consideration. Prediction change refers to the proportion of CLUEs classified differently than their corresponding original inputs. All values shown are averages across all testset points above the uncertainty rejection threshold.

**Applying CLUE to non-Bayesian NNs:** These models are unable to capture model uncertainty. We train deterministic NNs on every dataset under consideration using the architectures described in Appendix B.4. We generate counterfactuals for their noise uncertainty. As shown in Figure 24, CLUE is effective at explaining away noise uncertainty for regular NNs. More uncertain inputs are subject to larger changes in terms of $\ell_1$ distance.

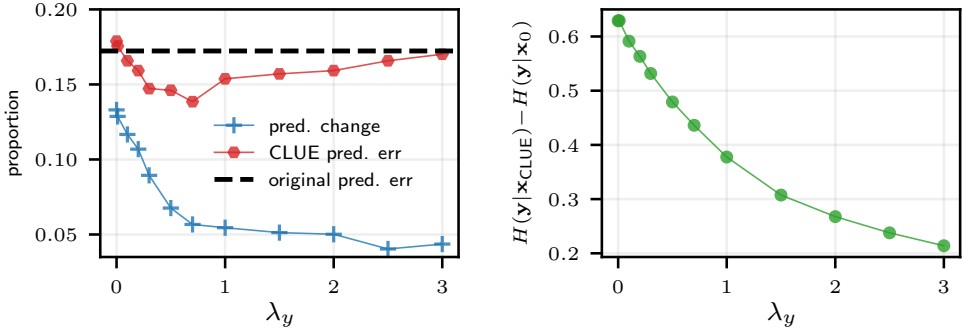

Figure 23: Left: Prediction change refers to the proportion of CLUEs classified differently than their corresponding original inputs. Setting a value of $\lambda_y$ of around 0.7 results in class predictions for CLUEs being closer to the true labels than the original class predictions. Right: Reduction in predictive entropy achieved by CLUE. All values shown are averages across all testset points above the uncertainty rejection threshold.

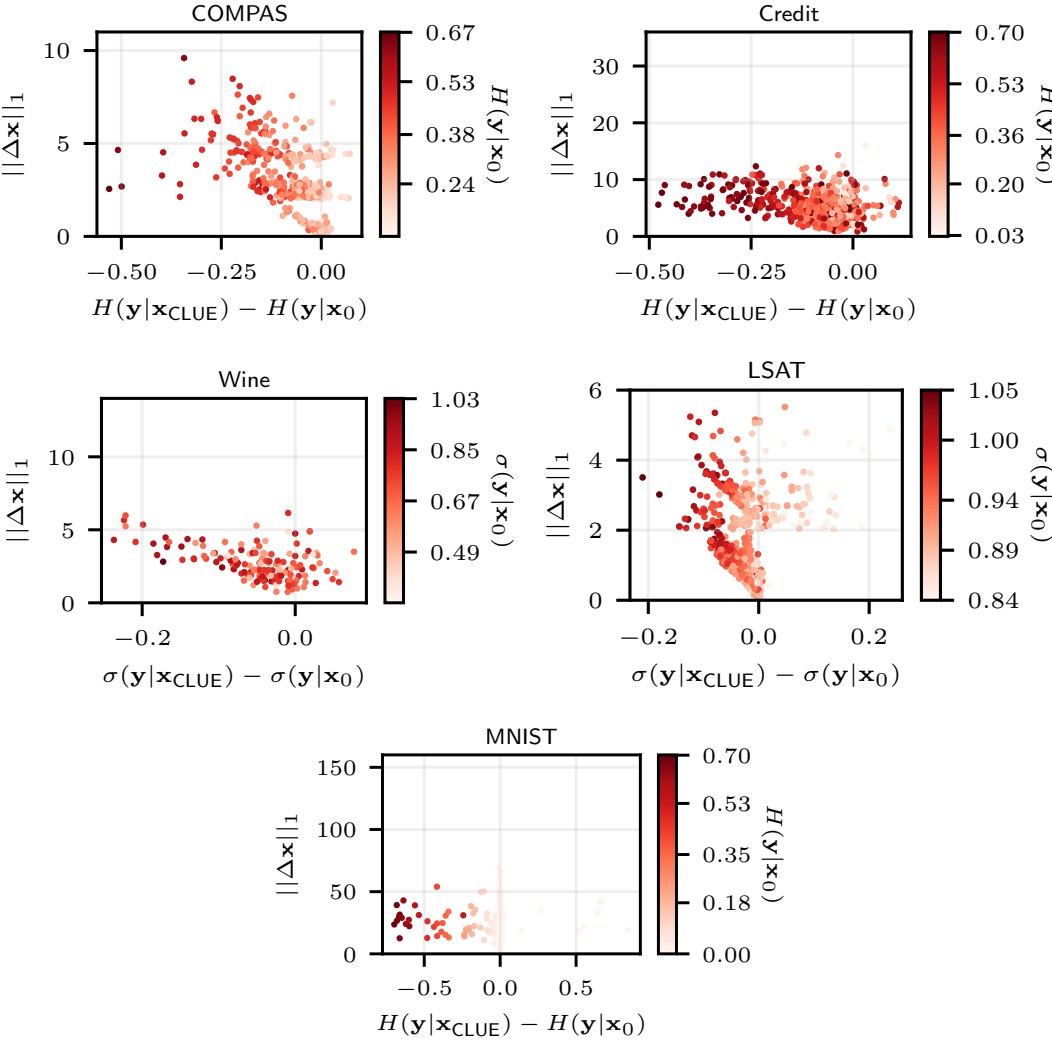

Figure 24: Amount of noise uncertainty explained away vs $\ell_1$ shift in input space for all datasets under consideration when applying CLUE to regular NNs. The colorbar indicates the original samples' predictive uncertainty.

## H.2 VERIFYING RESULTS FROM OUR COMPUTATIONAL EVALUATION FRAMEWORK

Our computational evaluation framework relies on generating artificial data. There is reasonable concern that the characteristics of this data may not reflect that of real-world data, biasing our results. As explained in Appendix I, we are careful to use powerful g.t. DGM models that generate high quality artificial data. Be that as it may, we validate the results from our computational evaluation framework by performing an analogous *informativeness* vs *relevance* experiment on real data.

As we do not have access to the generative process of the real data, we can not exactly quantify the uncertainty of inputs or how in-distribution they are. We instead resort to quantifying *informativeness* as the amount of our BNN's predictive uncertainty explained away $\Delta\mathcal{H} = \mathcal{H}(\mathbf{y}|\mathbf{x}_0) - \mathcal{H}(\mathbf{y}|\mathbf{x}_c)$. We measure *relevance* as the $L_2$ distance of each counterfactual to its $L_2$ nearest neighbor within the train-set $d_{NN-2}(\mathbf{x}_c, \mathcal{D})$.

We report mean values of $\Delta\mathcal{H}$, $d_{NN-2}(\mathbf{x}_c, \mathcal{D})$ and $\frac{\Delta\mathcal{H}}{d_{NN-2}(\mathbf{x}_c, \mathcal{D})}$ across all uncertain test points. A test point is deemed to be uncertain according to the criteria outlined in Appendix B.4. The hyperparameters employed for CLUE also match those provided in Appendix B.4. The step size used with local sensitivity analysis $\eta$ and U-FIDO's $\lambda_b$ parameter are found via grid search with a methodology analogous to the one described in Section 5.1.

Table 6: Quantities of *informativeness* ($\Delta\mathcal{H}$, higher is better), *relevance* ($d_{NN-2}(\mathbf{x}_c, \mathcal{D})$, lower is better) and their ratio ($\frac{\Delta\mathcal{H}}{d_{NN-2}(\mathbf{x}_c, \mathcal{D})}$, higher is better) obtained on real data from the LSAT, COMPAS and Wine datasets. The numbers in parenthesis indicate dataset dimensionality.

| Method | LSAT (4) | | | COMPAS (7) | | | Wine (11) | | |
|---|---|---|---|---|---|---|---|---|---|
| | $d_{NN-2}(\mathbf{x}_c, \mathbf{x}_0)$ | $\Delta\mathcal{H}$ | $\Delta\mathcal{H}/d_{NN-2}$ | $d_{NN-2}(\mathbf{x}_c, \mathbf{x}_0)$ | $\Delta\mathcal{H}$ | $\Delta\mathcal{H}/d_{NN-2}$ | $d_{NN-2}(\mathbf{x}_c, \mathbf{x}_0)$ | $\Delta\mathcal{H}$ | $\Delta\mathcal{H}/d_{NN-2}$ |
| Sensitivity | 0.482 | 0.003 | 0.0192 | 7.975 | 0.265 | 0.033 | 3.317 | 0.481 | 0.154 |
| CLUE | 0.080 | 0.092 | 1.664 | 0.067 | 0.014 | 0.737 | 1.274 | 1.409 | 1.188 |
| U-FIDO | 0.085 | 0.077 | 0.969 | 0.084 | 0.022 | 0.627 | 1.223 | 1.307 | 1.241 |

Table 7: Quantities of *informativeness* ($\Delta\mathcal{H}$, higher is better), *relevance* ($d_{NN-2}(\mathbf{x}_c, \mathcal{D})$, lower is better) and their ratio ($\frac{\Delta\mathcal{H}}{d_{NN-2}(\mathbf{x}_c, \mathcal{D})}$, higher is better) obtained on real data from the Credit and MNIST datasets. The numbers in parenthesis indicate dataset dimensionality.

| Method | Credit (23) | | | MNIST (784) | | |
|---|---|---|---|---|---|---|
| | $d_{NN-2}(\mathbf{x}_c, \mathbf{x}_0)$ | $\Delta\mathcal{H}$ | $\Delta\mathcal{H}/d_{NN-2}$ | $d_{NN-2}(\mathbf{x}_c, \mathbf{x}_0)$ | $\Delta\mathcal{H}$ | $\Delta\mathcal{H}/d_{NN-2}$ |
| Sensitivity | 0.770 | 0.224 | 0.121 | 6.903 | 0.601 | 0.087 |
| CLUE | 1.025 | 0.147 | 0.293 | 4.374 | 0.628 | 0.153 |
| U-FIDO | 1.863 | 0.017 | 0.052 | 4.887 | 0.409 | 0.088 |

As shown in Table 6 and Table 7, CLUE outperforms U-FIDO in terms of $d_{NN-2}$ on all datasets except wine, where both approaches are very similar. The same is true for the ratio $\frac{\Delta\mathcal{H}}{d_{NN-2}}$. Like in the artificial data experiments from Section 5.1, the difference between both methods is most stark for high dimensional datasets (MNIST and Credit). Here, CLUE is able to explain away more uncertainty while providing counterfactuals that are similarly close to the training data. Sensitivity is able to greatly reduce uncertainty in high dimensions. However, this comes at the cost going off the data manifold. In low dimensions, there are less possible directions in which steps can be taken, rendering the direct gradient based approach less powerful.

We note that the similarity of these results to the ones obtained in the analogous experiments from Section 5.1 suggest the unbiasedness of our computational evaluation framework.

## H.3 ADDITIONAL ANALYSIS OF USER STUDY

While the main text showed the mean accuracy of CLUE over all tabular questions, we also consider the breakdown of accuracy by dataset and by test point certainty in Table 8. CLUE outperforms all baselines on both datasets. We find that sensitivity does significantly worse in higher dimensions (on COMPAS), lending further credence to the intuition described in Appendix D.

When splitting by the certainty of test points, we immediately notice that accuracy for uncertain test points is quite high for all methods. This similarity is expected since certain context points are the only factor that varies between each method's survey. Survey participants seemed to not use the certain context points to identify uncertain test points. This is probably due to pilot procedure, wherein Participant A carefully paired test points with relevant uncertain context points. Indeed, the random baseline, which controls for the possibility that our task can be solved without access to a relevant counterfactual, performs best on uncertain test points. However, we note a large difference between methods' results when identifying certain test points. CLUE's accuracy almost doubles the second best method's (*Human CLUE*). When generating *Human CLUE*s Participant B had knowledge of the uncertain context point, but not the test point (just like other methods). For this reason, we expect to see dissimilarity in methods' performance on certain test points. CLUE's ability to bring about most relevant contrast is one possible explanation for why it does so much better than baselines for certain context points.

Table 8: Accuracy (%) of participants on the Tabular main survey broken down by dataset and by certainty of test points.

|  | Combined | LSAT | COMPAS | Certain Test | Uncertain Test |
|---|---|---|---|---|---|
| CLUE | **82.22** | **83.33** | **81.11** | **71.00** | 96.25 |
| *Human CLUE* | 62.22 | 61.11 | 63.33 | 38.00 | 92.50 |
| Random | 61.67 | 62.22 | 61.11 | 31.00 | **100** |
| Local Sensitivity | 52.78 | 56.67 | 48.89 | 20.90 | 92.50 |

# I  ADDITIONAL DETAILS ON THE GENERATIVE MODEL USED IN THE PROPOSED COMPUTATIONALLY GROUNDED EVALUATION FRAMEWORK

The framework described in Figure 6 uses a conditional DGM, specifically a VAEAC (Ivanov et al., 2019), to both generate artificial data and to evaluate explanations for said data. VAEs are known for generating blurry or overly smoothed data. For our evaluation framework to work well, we require the ground truth DGM to generate sharp data, with atypical characteristic that would to lead to a predictor being uncertain. We can ensure that this is the case by using a large latent dimensionality. However, this brings forth another well-known issue with VAEs: distribution mismatch (Dai & Wipf, 2019; Antoran & Miguel, 2019; Rosca et al., 2018). The region of latent space where the encoder places probability mass, also known as the aggregate posterior,

$$q_\phi(\mathbf{z}) = \int q_\phi(\mathbf{z}|\mathbf{x})p(\mathbf{x})\, d\mathbf{x}$$

does not match the prior $p(\mathbf{z})$.

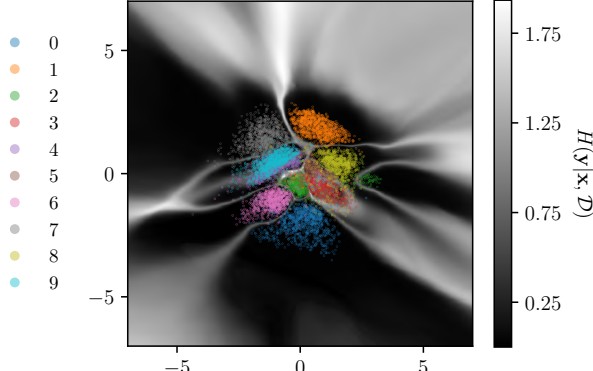

Figure 25: Predictive entropy estimates for artificial MNIST digits generated from a 2-dimensional VAE latent space. The MNIST test set digits have been projected onto the latent space and are displayed with a different color per class.

To visualize this phenomenon, we train a BNN and a VAE on MNIST. We sample points from the VAE's latent space and evaluate their uncertainty with the BNN. As shown in Figure 25, clusters of same-class digits form in latent space. The aggregate posterior presents low density in the spaces between clusters. Digits generated from these areas are of low-quality, causing our BNN to be uncertain. The outer regions of latent space, where the isotropic Gaussian prior has low density, also generate uncertain digits.

Recently, Dai & Wipf (2019) have proposed the two-level VAE as a solution to distribution mismatch. After training a standard VAE, a second VAE is trained on samples from the first VAE's latent space. As illustrated in Figure 26, the aggregate posterior over the inner latent variables, which we denote by $q(\mathbf{u})$, more closely resembles the prior. The joint distribution over inputs and latent variables factorizes as: $p(\mathbf{x}, \mathbf{z}, \mathbf{u}) = p(\mathbf{x}|\mathbf{z})p(\mathbf{z}|\mathbf{u})p(\mathbf{u})$. We refer the reader to (Dai & Wipf, 2019) for a detailed analysis. Figure 27 shows that, while generating digits from samples of $p(\mathbf{z})$ results in a large amount of low-quality or OOD reconstructions, samples from $p(\mathbf{u})$ map to clean digits. The two-stage mechanism restores the VAE's pivotal ancestral sampling capability, ensuring that our experiments with artificial data will be representative of methods performance on real data.

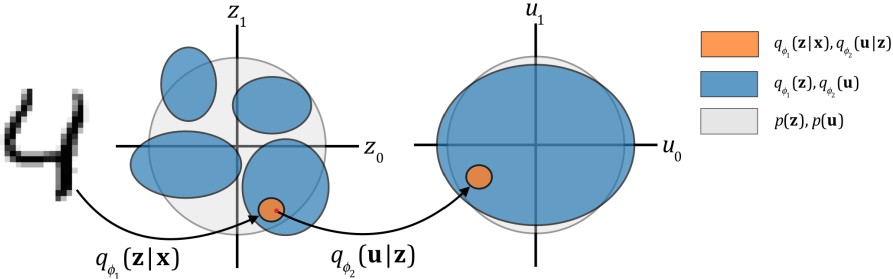

Figure 26: In its first stage, the two-level VAE maps input samples to approximate posteriors in the outer latent space. The aggregate posterior over this latent space need not resemble the isotropic Gaussian prior. The second VAE maps samples from the outer latent space to approximate posteriors in the inner latent space. The aggregate posterior over the inner latent space more closely matches the prior.

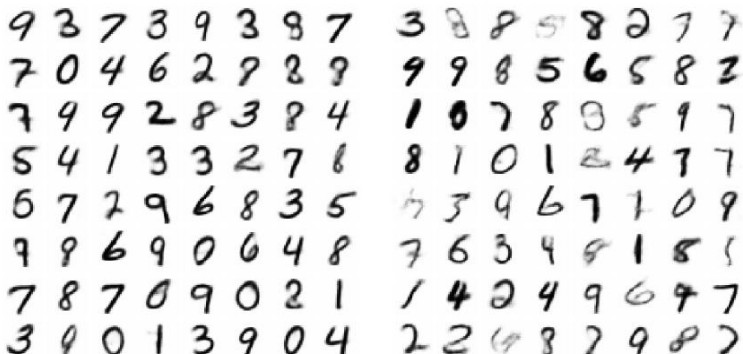

Figure 27: Left: Digits generated from the inner latent space of a VAEAC trained on MNIST with a two-level mechanism. Right: Digits generated from the latent space of a VAEAC trained on MNIST. $\mathbf{u}$ and $\mathbf{z}$ are drawn from $\mathcal{N}(\mathbf{0}, I)$.

In order to generate artificial data, we draw samples from the auxiliary latent space, map them back to the VAEAC's latent space and then map them to the input space. This allows for high-quality sample generation. In this way, a single VAEAC can be used for both ancestral sampling and conditional sampling. In addition, it allows us to estimate the log-likelihood of inputs as:

$$\log p_{gt}(\mathbf{x}) = \log \int p_{\theta_1}(\mathbf{x}|\mathbf{z})p_{\theta_2}(\mathbf{z}|\mathbf{u})p(\mathbf{u}) \, d\mathbf{z} \, d\mathbf{u} \qquad (8)$$

In (8) parameter subscripts refer to the outer (1st level) and inner (2nd level) networks. In order to preserve computational tractability, we approximate $p_{\theta_2}(\mathbf{z}|\mathbf{u})$ with a point estimate placed at its mean

$p_{\theta_2}(\mathbf{z}|\mathbf{u}) \approx \delta(\mathbf{z} - \mu_{\theta_2}(\mathbf{z}|\mathbf{u}))$. We further approximate (8) with importance sampling:

$$\log p_{gt}(\mathbf{x}) \approx \log \frac{1}{K} \sum_{k=1}^{K} \frac{p_{\theta_1}(\mathbf{x}|\mathbf{z}=\mu_{\theta_2}(\mathbf{z}|\mathbf{u}_k))p(\mathbf{u}_k)}{q(\mathbf{u}_k|\mathbf{x})}; \quad \mathbf{u}_k \sim q(\mathbf{u}|\mathbf{x}) \tag{9}$$

### I.1 Comparison of Methods under a Ground Truth DGM

The two-level VAEAC setup described above partially addresses the concern that our synthetic data might not be diverse enough to highlight differences among the methods being compared. Indeed, our results from Table 1 and Table 2 show noticeable differences in performance across methods.

We now address the opposite concern; methods that leverage auxiliary VAEs might be unfairly advantaged under our functionally grounded framework, as the generative process of our synthetic data is also VAE-based. Because VAEs are very flexible neural network based generative models, using them as a ground truth provides relatively little inductive biases for auxiliary DGMs to take advantage of. Additionally, our ground truth VAEAC captures the joint distribution of inputs and targets. The metric of interest, $\Delta\mathcal{H}_{\mathrm{gt}}$, only depends on the conditional distribution over targets $p_{gt}(\mathbf{y}|\mathbf{x})$. Our auxiliary DGMs only model inputs.

Two of the methods we evaluate, U-FIDO and CLUE, leverage auxiliary DGMs. Thus, both would be equally advantaged. The $\Delta\mathcal{H}_{\mathrm{gt}}$ vs $\Delta \log p_{\mathrm{gt}}$ metric from Table 2 is the most dependent on the ground truth VAEAC. However, we observe the largest difference between CLUE and U-FIDO on in this metric.

## J Details on User Study

### J.1 Additional Details on Tabular User Study

For our pilot point selection procedure, we take points from each dataset's test set that score above the uncertainty rejection thresholds described in Appendix B.4 as uncertain points. Points below the thresholds are labeled as certain points. Pilot procedure participants, referred to as participant A in the main text, were not informed that the pools were split up by the points' certainty with respect to the BNN being explained.

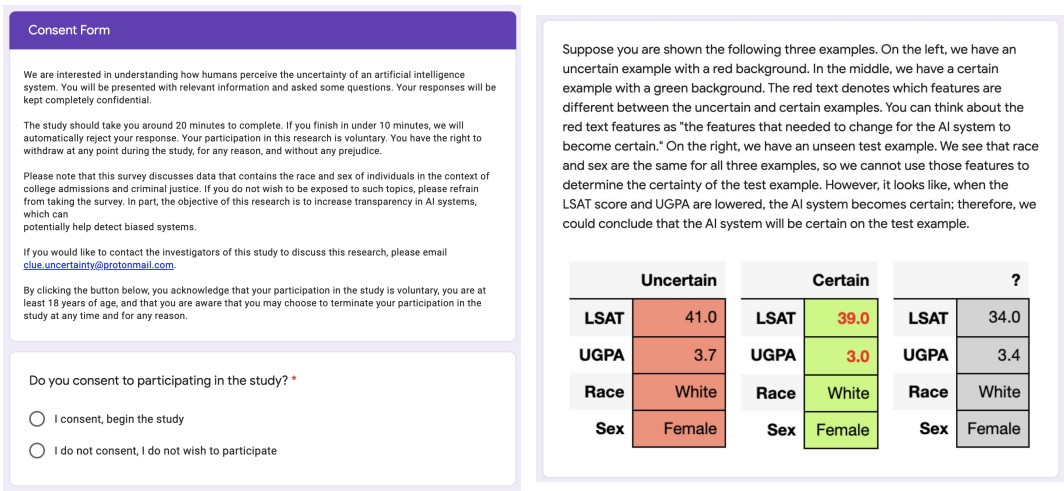

(a) Consent Form for the Tabular Main Survey    (b) Attention Check for the Tabular Main Survey

Figure 28: Setup of tabular user studies.

We now go through the various sections of the main survey. In Figure 28a, we include the consent form used in our user studies. This user study was performed with the approval of the University of Cambridge's Department of Engineering Research Ethics Committee. Only three participants who

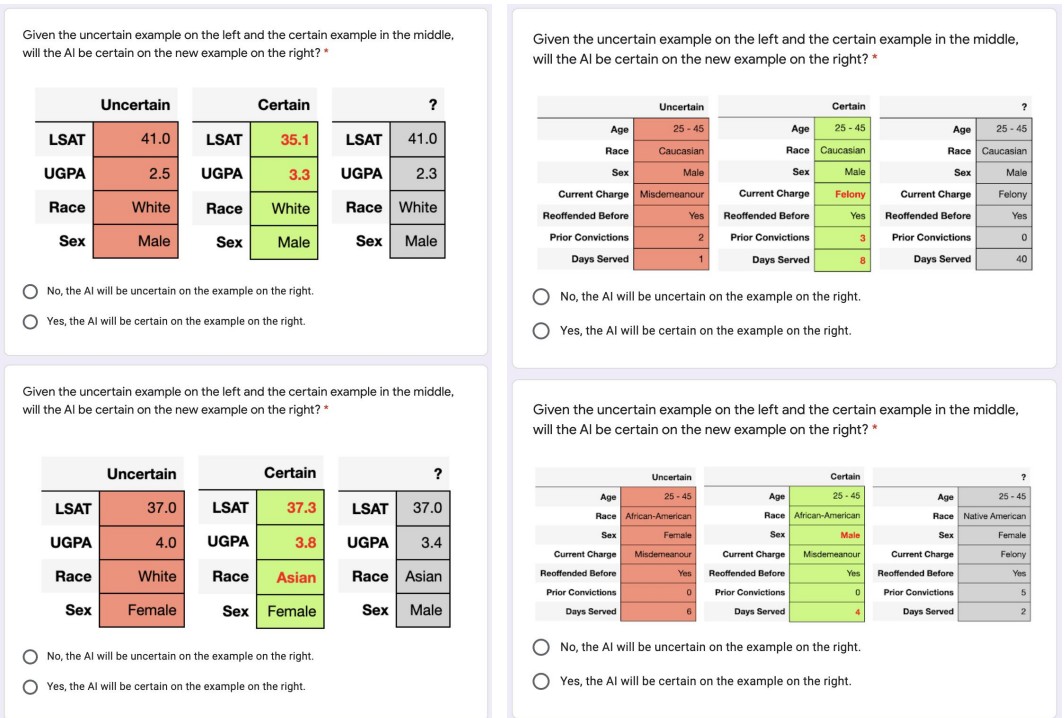

(a) Two LSAT questions with certain points generated by CLUE

(b) Two COMPAS questions with certain points generated by CLUE

Figure 29: Example Tabular Main Survey questions

were asked to take the survey did not provide consent and thus exited the form. We still ensured that at least ten participants took each of the four survey variants.

We then include an example question for each dataset, called an "attention check." An example is shown in Figure 28b. Note that the answer to this example question is provided in line. Later in the survey, we ask participants this exact same question. We ask one attention check per dataset. If participants get the attention check wrong for both datasets, we void their results. We only had to void one result. This did not affect our criteria of ten completed surveys per variant. The consent form and attention check questions were the same for all survey variants. The main survey participants were first asked the ten LSAT questions followed by the ten COMPAS questions: we made this design decision since the dimensionality of LSAT is lower than that of COMPAS, easing participants into the task. Examples of questions from the CLUE survey variant are shown in Figure 29.

## J.2 MNIST USER STUDY

In order to validate CLUE on image data, we create a modified MNIST dataset with clear failure modes for practitioners to identify. We first discard all classes except four, seven, and nine. We then manually identify forty sevens from the training set which have dashes crossing their stems. Using K-nearest-neighbors, we identify the twelve sevens closest to each of the ones manually selected. We delete these 520 sevens from our dataset. We repeat the same procedure for fours which have a closed, triangle-shaped top. We do not delete any digits from the test set. We train a BNN on this new dataset. Our BNN presents high epistemic uncertainty when tested on dashed sevens and closed fours as a consequence of the sparsity of these features in the train set.

We evaluate the test set of fours, sevens, and nines with our BNN. Datapoints that surpass our uncertainty threshold are selected as candidates to be shown in our user study as uncertain context examples or test questions. We show example CLUEs for a four and a seven that display the characteristics of interest in Figure 30.

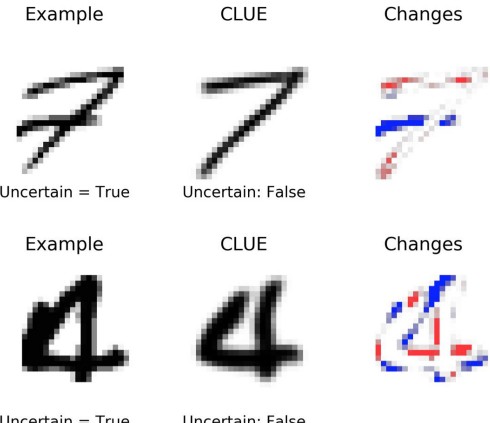

Figure 30: Examples of high uncertainty digits containing characteristics that are uncommon in our modified MNIST dataset. Their corresponding CLUEs and ΔCLUEs are displayed beside them.

Leveraging the modified MNIST dataset, we run another user study with 10 questions and two variants. Unlike our tabular experiments, we show practitioners a set of five *context points* to start, as opposed to a pair. This set of *context points* is chosen at random from the training set. The first variant involves showing users the set of *context points*, labeled with if their uncertainty surpasses our predefined threshold. We then ask users to predict if new test points will be certain or uncertain to the BNN. The second variant contains the same labeled context points and test datapoints. However, together with uncertain context points, practitioners are shown CLUEs of how the input features can be changed such that the BNN's uncertainty falls below the rejection threshold. The practitioners are then asked to decide if new points' predictions will be certain or not. If CLUE works as intended, practitioners taking the second variant should be able to identify points on which the BNN will be uncertain more accurately.

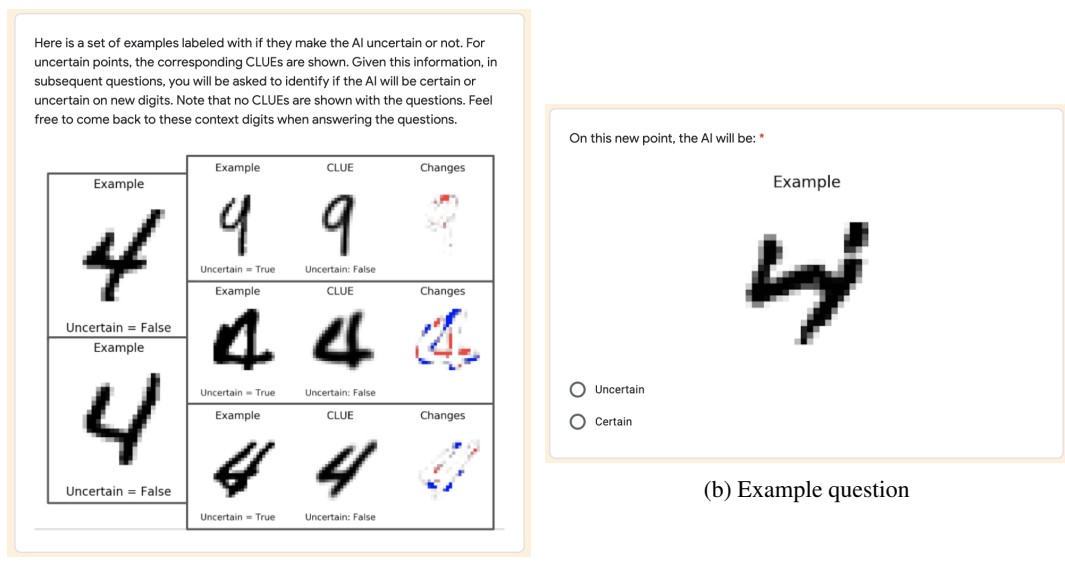

(a) Example Context Set with CLUEs

(b) Example question

Figure 31: MNIST User Study Setup

The first variant was shown to 5 graduate students with machine learning expertise who only received context points and rejection labels (uncertain or not). This group was able to correctly classify 67% of the new test points as high or low uncertainty. The second variant was shown to 5 other graduate students with machine learning expertise who received context points together with CLUEs in cases of high uncertainty. This group was able to reach an accuracy of 88% on new test points. This user study suggests CLUEs are useful for practitioners in image-based settings as well.

