# OpenReview forum: "Getting a CLUE: A  Method for Explaining Uncertainty Estimates"
_ICLR.cc/2021/Conference — ICLR 2021 Oral_

### Official Review · AnonReviewer2 · 2020-10-24

**Rating:** 6
**Confidence:** 4

**Review:**

This paper introduces CLUE -- a method to explain uncertainty estimates.  The method utilizes a VAE trained on the original data set to search effectively for low confidence instances.  The method utilizes a gradient based search through the latent space of the VAE.  The authors assess their approach on a variety of tabular data sets and MNIST.  They evaluate along change in uncertainty of the counterfactual as well as human evaluation.  They generally find improvements using their method over baselines.  Additionally, their method works much better in human evaluations.

Comments + Questions

Section 1:
- The authors argue that CLUE can be used to complement feature importance techniques like LIME, Saliency Maps, etc. They point out that when the model is confident, you can use a feature importance technique.  When it is not confident, you can use CLUE.  However, the motivation behind why this dual approach is useful is not quite clear.  With feature attribution methods, the goal is to understand what features the model is locally relying on.  If the goal is to understand how the model behaves and the model is uncertain for a particular point, it could still be quite insightful to use feature attribution methods.  The authors could better argue why their complementary technique is useful and make explicitly clear why you wouldn't want to use feature importance methods for uncertainty data instances like they suggest. Right now, it is not so clear.
- One minor point is that in the second paragraph, the authors motivate their method by saying CLUE can be useful to understand features contributing to uncertainty for instances underrepresented in the training data.  This motivation doesn't connect quite so clearly to the example in figure 1 --- we can see that the data instance is somewhat ambiguous.  Would the solution here to be to collect more ambiguous 6's? This motivating example could flow more clearly if it were a tabular instance because it would immediately connect to the motivating scenario given immediately before.

Section 3:
- In section 3, we see CLUE "aims to find points in latent space which generate inputs similar to an original observation x0 but are assigned low uncertainty."  However, in the introduction it was stated that “CLUEs answer the question: What is the smallest change that could be made to an input, while keeping it in distribution, so that our model becomes more certain in its decision for said input?” These two claims seem slightly at odds.  Should the claim in the introduction be revised?

Section 4:
- The evaluation procedure claims using data generated through a VAE as the training data will reduce the possibility of clue exploiting adversarial vulnerabilities.  Recent work has pointed out adversarial vulnerabilities might be part of the training data for image data sets as nonrobust features [1] and in this way could be captured by the VAE making this procedure less effective.  The bulk of this work is focused on tabular datasets and MNIST where this is less likely an issue, so I am not too concerned.  However, scaling this procedure up to larger image data sets could produce issues.
- A larger concern is that by only using data produced through a VAE, CLUE has a bit of an unfair advantage over methods like localized sensitivity which don't explicitly require the use of a VAE.  Meaning, the representational capacity of the VAE trained for CLUE is limited and in this way may not find certain diverse data points with low confidence.  A method that doesn't use a VAE might be able to find these points -- though search could be more challenging.  By forcing the set of images we're considering to be those produced by a VAE, this technique is shifting the playing field in favor of CLUE in what feels like a bit unfair way.  The evaluation should take into consideration that the requirement to train a VAE could be a disadvantage of CLUE but establish it is worthwhile nonetheless.

Section 5:
- From table 1, CLUE's performance seems relatively well balanced with U-FIDO in many of the tabular tasks.  The authors point out at the bottom of page 6 that CLUE performs better on higher dimensional data sets.  However, this is only apparent in the MNIST data set.  If CLUE's merits lie with higher dimensional data sets like images, it could be better to provide more evaluation in these settings.
-  In the reference for appendix h.2 and the $log p_{gt}(\cdot)$ test, it again feels like the assessment is a bit unfairly advantaged to CLUE; it feels very likely for CLUE to produce the best counterfactuals according to this metric because we assess data likelihood using a VAE.  At the same time, CLUE only generates counterfactuals on the VAE manifold.
- Maybe the authors could consider including some metric like nearest neighbor distance to the original training set for both clue and the baselines (where the baselines are run without being restricted to VAE generated data)? This could help us better understand the limitations imposed by using a VAE with CLUE. I think that an additional metric that disentangles the effect of the VAE within CLUE is needed here.
- The human study results add a lot of merit to the CLUE approach.  It's clear from these that the counterfactuals produced by clue are much more human interpretable.
- Though the right hand side of figure 10 helps use understand the limitations of using a VAE and is much appreciated, I still think an additional evaluation metric is needed for the 5.1 experiments.  An additional metric would help understand if the section 4 technique is giving CLUE an unfair advantage over the baselines.

Overall:
I am convinced after reading the paper that the method exhibits useful performance in finding human meaningful uncertainty focused counterfactuals.  Further, there are a number of strong experiments in the paper -- I particularly liked the human evaluation and found this convincing. That said, there are a number of weaknesses that I'll mainly divide into two categories:
(1) Introduction: per my comments in the introduction, I think this section could be significantly strengthened.  Most importantly here, the authors describe their method being used in an explanation workflow where uncertain instances are explained with clue and certain instances with something like LIME.  This thread, which seems like a key focus initially, is dropped for the rest of the paper.  It's currently unclear why this workflow makes sense and warrants much more justification.  Further, it could be worthwhile just to motivate CLUE as a method to explain uncertainty estimates in its own right because the connection with methods like LIME isn't explored in the rest of the paper.
(2) Evaluation technique from section 4: Using data generated from a VAEAC as the set of legitimate images could give CLUE an unfair advantage over baselines because it reduces the potential effects caused by representation capacity of the CLUE VAE. It would make this section much stronger to include another metric to try and isolate these effects.  I would appreciate some author clarification here as well, in case I am misunderstanding something about the evaluation technique.

One final minor point is the authors claim their method works better than baselines in higher dimensional data.  However, they only evaluate using MNIST where VAE's tend to be very strong.  To fully substantiate this claim, it could be worthwhile to consider a few slightly more challenging data sets for VAEs (street view house numbers, celeba, etc).

My sentiments are currently leaning towards reject mainly due to the motivation and experimental issues described in (1) and (2).  If the authors could remedy these concerns, I'd be inclined to raise my score because I think their are a number of potential valuable contributions in the work.

One related method that isn't discussed is https://arxiv.org/abs/2002.10248.  The authors generate instances at certain levels of prediction confidence --- though different it could be good to bring up.

[1] https://arxiv.org/abs/1905.02175

---- update ----
In response to the author's comments and extensions, I've raised my score.

---

> ### Author Response · Authors · 2020-11-19
> **Thank you for the feedback! (1/3)**
>
> We thank the reviewer for their thorough feedback and constructive suggestions. We address individual points below:
>
> ### Introduction: On the motivation behind CLUE and why feature attribution methods are unsuitable for uncertain inputs
>
> We agree that explaining uncertainty estimates is a worthwhile objective in its own right. We regret that this did not come through as clearly in our introduction as we would have liked. *We have modified the introduction to make it clear that the described workflow and Figure 1 are simply an example of a possible application of CLUE, not the main guiding motivation behind CLUE.*
>
> You raise a good question regarding the application of feature importance approaches to explaining uncertain inputs. Feature attributions will try to explain a models’ decision and not its uncertainty. When a model is not confident, its prediction will probably be wrong. The question is then, why would you want to explain a potentially wrong decision? When the model is not confident, the potentially wrong decision is likely to be the result of factors not related to the actual patterns or regularities present in the data. Examples of those factors are random initialization of weights, the order in which the training data was processed, prior assumptions, etc. Therefore, in this case, where the model is not confident, it makes much more sense to explain why the model is not confident than the actual model's decision.
>
> In Appendix F, we perform some preliminary experiments with feature importance techniques. We find, feature attribution methods fare poorly when presented with low confidence inputs. In these scenarios, we tend to get weak or noisy attributions, as there is often only weak evidence for each class. This problem is compounded when dealing with a classifier that captures model uncertainty, as lack of confidence could stem from multiple plausible weight configurations disagreeing with each other with individually high confidence.
>
> With the exception of FIDO, which relies on a generative model, the feature importance methods under consideration are difficult to retrofit for uncertainty. They are unable to add features. Instead, they are limited to explaining the contribution of existing features. This may suffice if our input contains all the information needed to make a certain prediction but otherwise results in noisy, potentially meaningless, explanations.
>
> Finally, feature importance approaches often require a choice of class to produce explanations. This complicates their use in scenarios where our model is uncertain and multiple classes have similar predictive probability. A counterfactual approach like CLUE is able to target uncertainty directly, providing a more meaningful local explanation. The following works also touch on the unreliability of saliency maps when a model’s weight are random (https://arxiv.org/abs/1810.03292) and when the data is OOD (https://arxiv.org/abs/2011.05429).  *We have further modified the introduction to better explain why feature attribution is typically not suitable for uncertain inputs.*
>
> ### Relationship of Example in Figure 1 with scenario where test are features underrepresented in training data.
>
> You are right, in the example from the introduction the source of uncertainty is class ambiguity (aleatoric uncertainty), not OOD features. There is evidence in the input for different classes (6 and 8). This figure is illustrative of our second motivating argument: directing the attention of the users of a data-driven decision making system to anomalous characteristics in the inputs.
>
> ### Discrepancy in phrasing of method between introduction and Section 3.
>
> For reference, in the introduction we state: “CLUEs answer the question: What is the smallest change that could be made to an input, while keeping it in distribution, so that our model becomes more certain in its decision for said input?” In Section 3: "CLUE aims to find points in latent space which generate inputs similar to an original observation $x_{0}$ but are assigned low uncertainty."
>
> We struggle to see a discrepancy in meaning of the provided descriptions. Could you please further elaborate? The phrasing used in section 3 is a bit more technical as the algorithm used to generate CLUEs is provided in that section. Please note that because points are generated by decoding from the latent space with an input similarity constraint (Eq 4), the resulting CLUEs will be in-distribution and will be similar in input space to x0. The phrasing in the introduction aims to provide a high level overview of the approach.

---

> > ### Author Response · Authors · 2020-11-19
> > **Thank you for the feedback! (2/3)**
> >
> > ### Evaluation Framework (Section 4): Potentially unfair evaluation when using VAE data generators
> >
> > We appreciate your concerns regarding the use of a VAE-based model as a generative process for artificial data. Thanks for suggesting the Nearest Neighbour distance to train set experiment. We agree it would help dispel concerns regarding our evaluation under a "ground truth" generative model. **Please see the comment above for this experiment's results and some discussion**.
> >
> > We carefully designed our functional experiments such that the result would not be impacted by our choice of “ground-truth” data generating model. This is discussed in Appendix I and summarised below.
> >
> > First of all, we would like to clarify that the generative model which acts as a ground truth generative process for artificial data is different from the generative models used by CLUE and U-FIDO. As described in Appendix I, we employ a 2-level VAE structure (https://arxiv.org/abs/1903.05789) for our ground truth DGMs, which increases these models’ expressiveness significantly. We train our ground truth DGMs on real data and then generate artificial data with said generative model. In the realm of artificial data, this generative model is the ground truth. Thus, its uncertainty represents the true aleatoric uncertainty of the generated data with respect to the generating process of the artificial data. This allows us to measure the effectiveness of different explanation methods exactly. In our experiments with artificial data, both CLUE’s and FIDO’s generative models are trained on artificial data from the ground truth generative model.
> >
> > In appendix I.1, we dispel two potential concerns with the above setup:
> >
> > The first potential issue is that the synthetic data might not be diverse enough to highlight differences among the methods being compared. The two-level VAEAC setup described above partially addresses this point. More importantly, our results from Table 1 and Table 5 show noticeable differences in performance across methods. This includes CLUE and U-FIDO, both of which use VAE-based generative models to build counterfactuals.
> >
> > We now address the opposite concern; methods that leverage auxiliary VAEs might be unfairly advantaged under our functionally grounded framework, because the generative process of our synthetic data is also VAE-based. Because VAEs are very flexible neural network based generative models, using them as a ground truth generative process provides relatively little inductive biases for auxiliary DGMs to take advantage of. Again, we do not observe hints of this sort of bias in our results for either experiment.
> >
> > 1) In our first experiment (Table 1), for each method, we compare the amount of uncertainty explained away by counterfactuals $\Delta H_{gt}$  with the distance between counterfactuals and original inputs $||x_{0} - x_{c}||$. Only $H_{gt}$ is measured with the ground-truth generative model. Critically, the metric of interest, $H_{gt}$, only depends on the ground truth VAEAC’s conditional distribution over targets $p_{gt}(y|x)$. U-FIDO and CLUE’s auxiliary DGMs only model inputs $p(x)$.
> >
> > 2) In our second experiment (Table 5), $||x_{0} - x_{c}||$ is substituted by counterfactual explanations’ probability density under the ground truth VAEAC  $p_{gt}(x_{c})$. Both our proposed method and baseline, (CLUE  and U-FIDO) leverage VAE-based auxiliary DGMs. Thus, both would be equally advantaged under this setup.  However, we observe larger differences in performance between CLUE and U-FIDO when using this metric than in the first experiment.
> >
> > *To further clarify this, we have added a reference to Appendix I from section 4.*
> >
> > Indeed, the results from the suggested experiment using real data and Nearest Neighbour distance resemble those from table 5 in our original manuscript.

---

> > > ### Author Response · Authors · 2020-11-19
> > > **Thank you for the feedback! (3/3)**
> > >
> > > ### Further concerns regarding generating counterfactuals with a generative model
> > >
> > > We see using an auxiliary generative model as a constraint which simplifies the problem of finding reasonable counterfactual explanations. Solving an unconstrained problem could, hypothetically, yield a more faithful solution but is intractably difficult in practise. This is easy to see from the poor performance of uncertainty sensitivity analysis in both functional experiments, user-study experiments and upon visual inspection (section 2.2, section 5 and Appendix D).
> > >
> > > We would like to note that if we removed the DGM from CLUE, the method would resemble a multistep version of uncertainty sensitivity analysis and would likely produce meaningless explanations. Indeed, most approaches to interpretability that involve counterfactual generation use generative models. (e.g. https://arxiv.org/pdf/1910.09398.pdf https://arxiv.org/pdf/1807.08024.pdf https://arxiv.org/pdf/1911.00483.pdf https://arxiv.org/pdf/1806.08867.pdf).
> > >
> > > Having said this, we share your concerns about the limitations imposed on this class of method by relying on generative models. This motivated the studies in section 5.3 as well as Appendices H.1 and I. The general takeaway from these is that CLUE’s generative model can be a bottleneck if it is not very expressive. However, using models with large latent spaces that are able to preserve a large part of the inputs’ information will not overly constrain the space of possible counterfactuals that can be generated. This can be qualitatively verified by observing appendix G: Here CLUE generates counterfactuals that preserve anomalous features in the input.
> > >
> > > ### On CLUE's similar performance to U-FIDO in table 1. and scaling to natural image datasets
> > >
> > > Table 1 examines the tradeoff between producing certain inputs and producing inputs that are close to the original sample being explained. Indeed here CLUE and U-FIDO perform similarly, although CLUE is clearly dominant on the small image dataset MNIST. In table 5, we perform an additional experiment where we compare the amount of uncertainty explained away with the probability density of counterfactual explanations. Here, we find that CLUE produces more in-distribution counterfactuals in 8 out of 10 settings. This can be seen as CLUE’s explanations being more likely to represent plausible data points, even when compared to U-FIDO which also relies on a generative model.
> > >
> > > We would like to clarify that CLUE is not restricted to using a VAE as its generative model. You are correct that VAEs generally fare poorly on very high dimensional data (although this is starting to change https://arxiv.org/abs/2007.03898). However, CLUE’s VAE could be readily swapped out with a GAN-based generative model. These are much stronger at modelling natural images. Our brief study in section 5.3 suggests that CLUE would work well with GAN-based generative models.
> > >
> > > ### Reference to  https://arxiv.org/abs/2002.10248
> > >
> > > We were not aware of this very recent work but it indeed seems very related to ours. *We have added a citation in our background section.* Thanks!

---

> > > > ### Comment · AnonReviewer2 · 2020-11-20
> > > > **Thank you for the detailed response and extensions**
> > > >
> > > > Thank you or the detailed responses and extensions. These experiments are convincing and have quelled my earlier concerns.
> > > >
> > > > In regards to the discrepancy point, I took a look at this again and realized I misunderstood what was being said.  I appreciate the authors clarification!
> > > >
> > > > As such, I've updated my score.

---

> ### Author Response · Authors · 2020-11-20
> **New Experiment with real data and Nearest Neighbour distance to train set (1/2)**
>
> ## New Experiment
> *In response to the concerns raised by AnonReviewer2 about evaluation on artificial data generated by a “ground truth” generative model:*
>
> We perform an analogous experiment on real data, where nearest neighbour distance to the train-set is used to measure the typicality of counterfactuals. For all methods under consideration, we compare the amount of uncertainty explained away  $\Delta \mathcal{H} = \mathcal{H}(y|x_{c}) - \mathcal{H}(y|x_{0})$ with the L2 nearest neighbour distance to the trainset  $d_{NN-2}(x_{c}, \cal{D})$ for each counterfactual example.
>
> We report mean values across all uncertain test points for $\Delta \mathcal{H}$, $d_{NN-2}$ and $\frac{\Delta \mathcal{H}}{d_{NN-2}}$. A test point is determined to be uncertain according to the criteria outlined in Appendix B.4. The $\lambda_{x}$ values used for CLUE are the same as in all other experiments. They can also be found in Appendix B.4. The step size used with local sensitivity analysis $\eta$ and U-FIDO’s $\lambda_{b}$ parameter are found via grid search with a methodology analogous to the one described in Section 5.1.
>
> **We will include the results from this experiment in the final manuscript.**
>
>
> | Method      | LSAT (4)                   |                      |                                       |   | COMPAS (7)                 |                      |                                       |   | Wine (11)                   |                       |                                       |
> |-------------|----------------------------|----------------------|---------------------------------------|---|----------------------------|----------------------|---------------------------------------|---|-----------------------------|-----------------------|---------------------------------------|
> |             | $d_{NN-2}(x_{c}, \cal{D})$ | $\Delta \mathcal{H}$ | $\frac{\Delta \mathcal{H}}{d_{NN-2}}$ |   | $d_{NN-2}(x_{c}, \cal{D})$ | $\Delta \mathcal{H}$ | $\frac{\Delta \mathcal{H}}{d_{NN-2}}$ |   | $d_{NN-2}(x_{c},  \cal{D})$ | $\Delta  \mathcal{H}$ | $\frac{\Delta \mathcal{H}}{d_{NN-2}}$ |
> | Sensitivity | 0.482                      | 0.003                | 0.0192                                |   | 7.975                      | 0.265                | 0.033                                 |   | 3.317                       | 0.481                 | 0.154                                 |
> | CLUE        | 0.080                      | 0.092                | 1.664                                 |   | 0.067                      | 0.014                | 0.737                                 |   | 1.274                       | 1.409                 | 1.188                                 |
> | U-FIDO      | 0.085                      | 0.077                | 0.969                                 |   | 0.084                      | 0.022                | 0.627                                 |   | 1.223                       | 1.307                 | 1.241                                 |

---

> > ### Author Response · Authors · 2020-11-20
> > **New Experiment with real data and Nearest Neighbour distance to train set (2/2)**
> >
> > | Credit (23)                 |                       |                                       | MNIST (784)               |                       |                                       |
> > |-----------------------------|-----------------------|---------------------------------------|---------------------------|-----------------------|---------------------------------------|
> > | $d_{NN-2}(x_{c},  \cal{D})$ | $\Delta  \mathcal{H}$ | $\frac{\Delta \mathcal{H}}{d_{NN-2}}$ | $d_{NN-2}(x_{c},\cal{D})$ | $\Delta  \mathcal{H}$ | $\frac{\Delta \mathcal{H}}{d_{NN-2}}$ |
> > | 0.770                       | 0.224                 | 0.121                                 | 6.903                     | 0.601                 | 0.087                                 |
> > | 1.025                       | 0.147                 | 0.293                                 | 4.374                     | 0.628                 | 0.153                                 |
> > | 1.863                       | 0.017                 | 0.052                                 | 4.887                     | 0.409                 | 0.088                                 |
> >
> >
> > CLUE outperforms U-FIDO in terms of $d_{NN-2}$ on all datasets except wine, where both approaches are very similar.  The same is true for the ratio $\frac{\Delta \mathcal{H}}{d_{NN-2}}$. Like in the artificial data experiments, the difference between both methods is most stark in high dimensional datasets (MNIST and Credit). Here, CLUE is able to explain away more uncertainty while providing counterfactuals that are similarly close to the training data.
> >
> > Sensitivity is able to greatly reduce uncertainty in high dimensions. However this comes at the cost going off the training manifold. In low dimensions, there are less possible directions in which steps can be taken, rendering the direct gradient based approach less powerful.
> >
> > We would like to note that these results are similar to the ones obtained in the analogous experiment with artificial data (table 5).
> >
> > We are open to additional feedback and suggestions.

---

### Official Review · AnonReviewer1 · 2020-10-25
**Simple, effective approach with strange framing**

**Rating:** 7
**Confidence:** 3

**Review:**

This paper addresses the problem of explaining the uncertainty of a prediction made by a differentiable probabilistic model (as opposed to the prediction itself) through counterfactual explanations. They propose a technique, CLUE, which optimizes their counterfactual metric in the latent space of a deep generative model. To validate their approach, they introduce a quantitative evaluation for uncertainty metrics and conduct a user study, while also analysing CLUE's reliance on the auxiliary DGM.

Strengths:
The empirical validation of the approach is strong.

The authors deserve particular credit for "creating their own baseline" by adapting a previous counterfactual approach (FIDO) to the problem at hand.

The user study is well-executed, and produces a pretty stark improvement over baselines, including human-selected counterfactuals.

While the introduced approach does require the non-trivial complexity of training a DGM, conceptually the method is an elegant way of dealing with one of the big challenges for counterfactual explanations - staying on the data manifold.

Weaknesses:
1. The framing (abstract/introduction) of the paper took a while to wrap my head around, and could probably be improved. In particular, for classification problems, the "simple, stupid" approach of looking for the most negative feature attributions for a prediction seems like it would produce an explanation of uncertainty (though not a counterfactual one, so not competitive to CLUE). This makes Figure 1 pretty puzzling, as it's not clear to me why standard attributions aren't useful for uncertain predictions.
2. Qualitatively, how would CLUE compare to a standard counterfactual explanation?
3. Why isn't U-FIDO included in the user study? Given that it performed the best in section 5.1 on the datasets used, the results would be interesting.
3. In the user study, is the test example linked to the two context points at all? I wouldn't expect unrelated context points to be that useful in classifying a random test example.

Nitpicks:
- In the appendix, CLUE is compared against Shapley/LIME on MNIST. LIME is a pretty strange choice, and has never been shown/claimed to be remotely SOTA on neural networks. Something like integrated gradients would be more relevant/interesting.

---

> ### Author Response · Authors · 2020-11-19
> **Thanks for the feedback!  (1/2)**
>
> We thank the reviewer for their encouraging words and helpful suggestions. We address individual points below:
>
>
> ### Confusion about intro: Why not just use negative feature attribution for uncertain inputs?
>
> You raise a good question. Feature attributions will try to explain a model's decision and not its uncertainty. When a model is not confident, its prediction will probably be wrong. The question is then, why would you want to explain a potentially wrong decision? Furthermore, when the model is not confident, the potentially wrong decision is likely to be the result of factors not related to the actual patterns or regularities present in the data. Examples of those factors are random initialization of weights, the order in which the training data was processed, prior assumptions, etc. Therefore, in this case, where the model is not confident, it makes more sense to explain why the model is not confident than the actual model's decision.
>
> In Appendix F, we perform some preliminary experiments with feature importance techniques. We find feature attribution methods fare poorly when presented with low confidence inputs. In these scenarios we tend to get weak or noisy attributions, as there is often only weak evidence for each class. This problem is compounded when dealing with a classifier that captures model uncertainty, as lack of confidence could stem from multiple plausible weight configurations disagreeing with each other with individually high confidence.
>
> Furthermore, with the exception of FIDO, which relies on a generative model, the feature importance methods under consideration are difficult to retrofit for uncertainty. They are unable to add features. Instead, they are limited to explaining the contribution of existing features. This may suffice if our input contains all the information needed to make a certain prediction but otherwise results in noisy, potentially meaningless, explanations.
>
> Finally, feature importance approaches often require a choice of class to produce explanations. This complicates their use in scenarios where our model is uncertain and multiple classes have similar predictive probability. A counterfactual approach like CLUE is able to target uncertainty directly providing a more meaningful local explanation.  The following works also touch on the unreliability of saliency maps when a model’s weights are random (https://arxiv.org/abs/1810.03292) and when the data is OOD (https://arxiv.org/abs/2011.05429).
>
> *We modified the introduction to present our method in a more stand-alone fashion and less in contrast with existing feature importance approaches. We also better explain why feature attribution is typically not suitable for uncertain inputs.*
>
> ### How would CLUE compare with a regular counterfactual explanation?
>
> Initially, we considered standard counterfactual generation approaches for explaining uncertainty. However, upon further consideration, we came to the conclusion that existing approaches would not be very well suited for the task. Counterfactual explanations can be seen as a type of approach to obtain feature importance (see https://arxiv.org/abs/2011.04917  for a unification of counterfactual explanations and feature importance). Thus we would expect regular counterfactuals to suffer from the pathologies described above when dealing with uncertain inputs. We elaborate below:
>
> As explained in https://arxiv.org/pdf/1807.08024.pdf, broadly speaking, a counterfactual explanation can be built in one of two ways:
>
> 1) “Smallest Deletion Region (SDR) considers a saliency map as an answer to the question: What is the smallest input region that could be removed and swapped with alternative reference values in order to minimize the classification score?”
>
> When applied to an uncertain input, we would not expect anything to happen as the input is already uncertain. Note that CLUE can be thought of an inversion of SDR: we search for the smallest perturbation necessary to make our input certain.
>
> 2) “Smallest Supporting Region (SSR) instead poses the question: What is the smallest input region that could substituted into a fixed reference input in order to maximize the classification score?”
>
> In cases where uncertainty stems from an input containing evidence for multiple classes, this approach could potentially work well. In situations where uncertainty stems from a lack of evidence for any classes, we would expect this approach to return spurious outputs.
>
> For the above reasons, we adapted the state of the art counterfactual generation approach (FIDO) to uncertainty for use as a baseline, instead of applying it in its original form.

---

> > ### Author Response · Authors · 2020-11-19
> > **Thanks for the feedback! (2/2)**
> >
> > ### Why isn't U-FIDO in the user study?
> >
> > Our access to users (4th year or Masters level Machine Learning students) is decently limited. This complicates running as many user study experiments as we would have liked. We decided to pick Uncertainty Sensitivity Analysis instead of U-FIDO for our user study, since Uncertainty Sensitivity Analysis is the only existing ML interpretability method from the literature that targets uncertainty (recall we had to retrofit FIDO to become U-FIDO to compare to CLUE). We also include the very strong human baseline, where counterfactual explanations are manually selected by other users.
> >
> >
> > ### "In the user study, is the test example linked to the two context points at all? I wouldn't expect unrelated context points to be that useful in classifying a random test example."
> >
> > This is an important point. Test data points are related to uncertain context data points. The selection of these is as follows: A participant who will *not* take the main study is shown a pool of certain and uncertain points and a separate pool of uncertain points. They are told to select pairs of related points, one from each pool. The points selected from the first pool will act as test points while the points selected from the second pool act as uncertain context points. We then apply our explanation methods to the uncertain context points to generate counterfactual explanations. Because the **test and uncertain context points are selected together**, we expect there to be some transferability between the counterfactual explanations for uncertain points and the test points.
> >
> > The above is described in the first two paragraphs of section 5.2 and in Figure 8.
> >
> > ### In appendix F: LIME is bad for explaining NNs, integrated gradients could be used instead.
> >
> > As discussed in Appendix F, we tried to apply Deep SHAP, a feature importance approach similar to Integrated Gradients in that it takes into account the internal structure of our classifier NNs by evaluating their gradients. Unfortunately, this method produced very noisy explanations when applied to our BNNs. We conjecture that this high variance might be induced by disagreement among the multiple weight configurations from our BNNs.
> >
> > We found approaches that treat the model as a complete black box to perform better. For this reason, we chose Lime and Kernel SHAP as baselines in the appendix and a FIDO based method as a baseline in the main text. You are of course correct that linear explanations, like LIME, of a non-linear model, like a NN, will necessarily be unfaithful. We discuss this in Section 2.2. We never claim LIME to be state of the art for NNs. However, we did find it to qualitatively outperform gradient-aware methods for explaining our BNNs’ uncertainty.
> >
> > We believe that the application of feature importance techniques to models that capture weight uncertainty is a very interesting direction for future research but it is outside of the scope of this work.

---

> > > ### Comment · AnonReviewer1 · 2020-11-24
> > > **Thanks for the detailed response**
> > >
> > > Thanks to the authors for their detailed response. My concerns have been essentially addressed.
> > >
> > > The problem of explaining uncertainty is, frankly, a little hard to wrap my head around (despite having worked in XAI for a number of years). The new introduction is good enough to be published (though still not great), and re-visiting the paper makes me more convinced that explaining uncertainty should be "a thing" that is worth breaking ground on. I'm also inclined to give the authors the benefit of the doubt for tackling a new problem, rather than producing the umpteenth feature attribution/counterfactual paper.
> > >
> > > I debated raising my score to 8, but am going to stay at 7 (for context, I was debating between 6 and 7 in my initial review, so this is really an increase of one point from 6.5 to 7.5 :) ).

---

### Official Review · AnonReviewer4 · 2020-10-28
**Sound, well-written paper, interpretability and uncertainty of models, perhaps most contribution to XAI community.**

**Rating:** 7
**Confidence:** 3

**Review:**

This paper tackles the problem of making Ai/ML-systems more trustworthy making the uncertainty associated with a model, in this case BNN, visible, more interpretable. Interpretability and knowing the limitations and uncertainties associated with a model are definitely very interesting research challenges. These topics are very relevant for ML, AI, Explainable AI etc., but I still think that they are also for ICLR (even if many conferences in the ML/AI/XAI will also fit this paper)

I find the main ideas innovative, the paper is well-written, explained and even includes some kind of “small” user study. The authors also provide a framework for evaluating the counterfactual explanations of uncertainty provided, using informativeness, and they carry out well-designed experiments for validating CLUE.

P. 7, under section 5.2. Can the first sentence be referred to Hoffman? I don’t think so. Are the participants used in the user study be good representatives of the possible users/practitioners of CLUE (as also stated in the conclusions)?

There has come a recent survey on counterfactuals, that I think it is relevant for this work:

Verma, S., Dickerson, J., & Hines, K. (2020). Counterfactual Explanations for Machine Learning: A Review. arXiv preprint arXiv:2010.10596.

I understand that it is out of the scope of this paper, but the notion of counterfactuals used in the ML community, like the one used in this paper (section 2.3), is quite narrow compared to how we use counterfactuals and contrastive explanations in real life. I think the richness and complexity of counterfactual explanations is well illustrated in

Byrne, R. M. (2019, August). Counterfactuals in Explainable Artificial Intelligence (XAI): Evidence from Human Reasoning. In IJCAI (pp. 6276-6282).

Perhaps this is something to discuss in the future.

(just to make clear: I am not involved in any of the references given, just thought that they can be of interest for this paper).

---

> ### Author Response · Authors · 2020-11-19
> **Thanks for the feedback!**
>
> We thank the reviewer for their encouraging words and insightful suggestions. We address individual points below:
>
> ### Reference to Hoffman et. al. in our user study (Sec 5.2)
>
> We think the citation is generally relevant to the construction of user studies, even though our study doesn't apply all of the techniques proposed in that paper. Could you please elaborate on why you do not see it as appropriate? We are not opposed to removing the citation if you feel strongly about it.
>
> ### Are the user study subjects representative of real users?
>
> In our introduction, we suggest CLUE could be useful to ML practitioners developing data-driven decision making systems and to domain experts working in conjunction with these tools. Our human subject experiment focuses on the former setting.
>
> Because our access to real-world ML practitioners is limited, our participants are Master students in Machine Learning. These students are a good proxy for ML practitioners as they will likely go on to become practitioners in the following years. We clarify our wording regarding this in the updated manuscript.
>
> ### Contrastive explanations, additional citations
>
> You bring up a good point: We placed a lot of emphasis on distinguishing counterfactuals in the causal inference sense from counterfactuals in explainability. However, we did not link these to the broader field of contrastive explanations. Our revised draft includes an additional comment in section 2.3 relating counterfactuals to contrastive explanations together with references to the works you mention.

---

> > ### Comment · AnonReviewer4 · 2020-11-19
> > **About Hoffman...**
> >
> > Hi authors,
> >
> > Thanks for your response. I think you have a very solid paper, and I only had minor comments, including the Hoffman citation comment, but I agree that I was not so clear. The reference is absolutely relevant in this context, but I had a problem regarding the sentence itself. The sentence was "CLUE’s promising results in computational evaluation do not substitute for human-based evaluation (Hoffman et al., 2018)"... I don't know what Hoffman reference adds there, do you mean that in general Hoffman et al. says that computational evaluation does not substitute for human-based evaluation (the second part of the sentence)? Is it not better to change the sentence so you make the point that human-based evaluations are needed (cite Hoffman) and then here comes an evaluation for CLUE?
> >
> > As I said, this was really just a minor comment...

---

> > > ### Author Response · Authors · 2020-11-19
> > > **Thanks for the clarification!**
> > >
> > > Yes, thank you, we'll amend the text as you suggest to be clearer.

---

### Official Review · AnonReviewer3 · 2020-10-29
**Review for Getting A CLUE: A Method for Explaining Uncertainty Estimates**

**Rating:** 7
**Confidence:** 4

**Review:**

## Summary

The authors consider the problem of post-hoc explainability for decisions rendered by machine learning models.  They focus on addressing uncertain model predictions, producing counterfactual data that is both likely under a generative model of the data, as well as more certain in the classification task.  They present both experimental evidence, as well as a user study geared towards practitioners, that show the benefits of counterfactual explanations targeting uncertainty.

## Strong points

What really sets this paper aside for me is its explicit focus on uncertain model decisions, as well as its inclusion of a user study.

- While other works do use the point estimates of model softmax outputs as part of their methods (e.g Progressive Exaggeration), they do not focus on providing counterfactual explanations specifically for those points where the model is uncertain.

- Most works in explainable AI, even those targeted towards practitioners, do not evaluate the utility of their tools in front of an audience of machine learning model developers.  Bravo for undertaking this work which is (currently) under-valued in the ML community.

## Weak points

- In step 4 of Algorithm 1, why do you need to focus on a BNN for obtaining $H(y|x)$?  Going back to my previous point about limiting usefulness of CLUE, there are many methods that yield $P(y | x)$, even by approximate Bayesian inference.

- Is it the case that as the method converges (if it converges?), $H(y|x)$ is likely to decrease, while $d(x, x_0)$ is likely to increase?  Do you attempt to anneal the relative contribution to the loss to account for this?  Furthermore, do you ensure that the relative contribution to the loss is balanced between $d(.,.)$ and $H(y|x)$??  If these become imbalanced, one will surely dominate the direction of $\nabla_Z \mathcal{L}$

- In the description of the baselines used for experiments in section 5, the localized uncertainty sensitivity analysis seems artificially weak; why not include a more robust ensemble of models that produce softmax output over class assignments?  Or a proper probabilistic model like a GP?

## Recommendation

The effectiveness of CLUE, and indeed of every counterfactual explanation method cited that makes use of an auxiliary generative model of the data is bounded by the faithfulness of this DGM to model the density.  This is a more fundamental limit on the applicability of these methods, not specific to CLUE, but worth stating IMO.

I disagree with the statement detailed in section 4 after the evaluation procedure involving $\mathcal{H}_{gt}$ capturing ground truth aleatoric uncertainty: $p_gt(y|x)$ is just another generative model trained by estimation of the data, it’s not special.  A better measure of aleatoric uncertainty would involve the variance of $p(y | x)$ taken over multiple independent models (see Snoek et al.).  I also find the argument about adversarial weakness that immediately follows is a bit confusing "*Approaches that exploit adversarial weaknesses in the BNN will not transfer to the g.t. VAEAC, failing to reduce uncertainty on error*".

Fundamentally, though I see areas where the work could be improved, I believe the work is sufficiently different to existing counterfactual explanation methods to be accepted.

## Questions for the authors

- In section 3 where you penalize the distance $d(x, x_0)$, given that the motivation of having a penalty on $d(x, x_0)$ is to try and ensure minimal changes, what about instead ensuring this by doing projected gradient onto the space of plausible data?

- Again in section 3, is $d_y(f(x), f(x_0))$ intended to be high, or low?  Traditional understanding of counterfactual explanations in the literature would suggest $f(x) != f(x_0)$, but I can see the value in not caring about enforcing this to focus on driving down $H(y | x)$.  Could you spend some space touching upon this design decision here?

- In Section 5.1, I'm curious why FIDO was chosen as a baseline?  If memory serves, the FIDO objective has additional constraints to try and ensure the B form a contiguous set, which your U-FIDO formulation (equations 6,7) does not admit.

## Suggestions for improving the paper

-  Algorithm 1 presents a minor nomenclature issue: the output $x_{clue}$ produced might not be an actual counterfactual in the sense of Wachter et al., in that no effort is made to orient the latent space edits (z) towards crossing a decision boundary for Y.

- At the beginning of section 4, I find step 4 of the procedure is not clearly presented.  Step 4 suggests it’s used as a way to measure whether the discovered x_c are likely given the density of the model.  Is that so? If not, could you be more clear why evaluation of $\tilde{x_c}$ by the VAEAC is helpful?

- At the end of section 3, it would be helpful here to consider recent efforts to encourage counterfactual explanations to be confined to small contiguous regions (cf. Dabkowski and Gal 2017, Chang et al 2019).  This issue of potentially large, disparate, sparse signals was a flaw of original gradient based saliency maps, and would be well addressed here.

- In section 5.1 where you discuss the criteria for evaluation of counterfactuals (*We would like counterfactuals to explain away as much uncertainty as possible while staying as close to the original inputs as possible*).  This is achieved, albeit indirectly, by other counterfactual generation methods (e.g Progressive Exaggeration https://openreview.net/forum?id=H1xFWgrFPS), which vary latent representations along a continuum of class output probabilities.  You could use their method as a comparator by selecting uncertain points and generating counterfactuals that move away from the decision boundary instead of towards it.

---

> ### Author Response · Authors · 2020-11-19
> **Thank you for the feedback! (1/2)**
>
> We thank the reviewer for their constructive feedback and detailed suggestions. We address individual points below:
>
> ### Why refer to “BNN” in algorithm 1 instead of a more general probabilistic predictive model?
>
> Good Point. Indeed, our method can be generally applied to any probabilistic model. We have clarified this in algorithm 1 and other parts of the paper. As stated in our introduction, our experiments focus on BNNs as they are an increasingly popular choice of model in Bayesian machine learning due to their flexibility and scalability to large amounts of data.
>
> ### On the relative contribution of H and d in objective.
>
> You are correct: as $H$ decreases, $d$ tends to increase. This is intended behaviour. It ensures that the algorithm converges to a minmax saddlepoint where counterfactuals are certain but not too different from original inputs. This also allows $d_{x}(x, x_{0})$ to generally be larger for more uncertain inputs (larger $H$). This can be seen in Figure 10 of the main text or Figure 20 of the appendix.
>
> It is indeed important to maintain balance between the contributions of both terms to the total loss. Otherwise, our algorithm might change the original input too much, or not do anything at all. In practise, we achieve this balance through the choice of the $\lambda_{x}$ hyperparameter (all values provided in appendix B.4). As described therein, we scale $\lambda_{x}$ based on the dimensionality of the input space $D$ in order to make the distance contribution to the loss $\lambda_{x}d_{x}(x, x_{0})$ agnostic to $D$. However, this does not completely compensate for the particular characteristics of each dataset’s input distribution so we finetune $\lambda_{x}$ via cross-validation.
>
> ### Weakness of Localized sensitivity analysis as a baseline.
>
> You are correct in that localized sensitivity analysis is not a strong baseline, as also discussed in section 2.2. Its inclusion is due to uncertainty sensitivity analysis being the only method in the literature aimed towards increasing the transparency of uncertainty estimates. We would highlight that we also adapt an existing state-of-the-art counterfactual explainability method (FIDO) to explain uncertainty estimates and use it as a baseline in our functional experiments. In our user study, we include the very strong human baseline, where counterfactual explanations are manually selected by other users.
>
> We might not have fully understood the second part of your comment here: "why not include a more robust ensemble of models that produce softmax output over class assignments? Or a proper probabilistic model like a GP?". Could you please further elaborate? -- In our experiments, all of our baselines are tasked with explaining the uncertainty estimates provided by the same BNN. What differs is the explanation method, not the model being explained. The model being explained could indeed have been an ensemble or a GP. This is further clarified in our updated document.
>
> ### Counterfactual explanation approaches could be limited by the faithfulness of a generative model.
>
> We use an auxiliary generative model as a constraint which simplifies the problem of finding reasonable counterfactual explanations. Solving an unconstrained problem could, hypothetically, yield a more faithful solution but is intractably difficult in practise. This is easy to see from the poor results obtained by uncertainty sensitivity analysis in section 2.2, section 5 and Appendix D. We would like to note that if we removed the DGM from CLUE, the method would resemble a multistep version of uncertainty sensitivity analysis and would likely produce meaningless explanations. Indeed, most approaches to interpretability that involve counterfactual generation use generative models. (e.g. https://arxiv.org/pdf/1910.09398.pdf https://arxiv.org/pdf/1807.08024.pdf https://arxiv.org/pdf/1911.00483.pdf https://arxiv.org/pdf/1806.08867.pdf). Fortunately, recent developments in generative models allow us to capture very complex input space distributions. An example would be the Projection GAN used by the authors in the progressive exaggeration paper you mention in your review.
>
> Having said this, we share your concerns about the limitations imposed on this class of method by relying on generative models. This motivated the studies in section 5.3 as well as Appendices H.1 and I. The general takeaway from these is that CLUE’s generative model can be a bottleneck if it is not very expressive. However, using VAEs with large latent spaces that are able to preserve a large part of inputs’ information will not overly constrain the space of possible counterfactuals that can be generated. This can be qualitatively verified by observing appendix G: Here CLUE generates counterfactuals that preserve anomalous features in the input.

---

> > ### Author Response · Authors · 2020-11-19
> > **Thank you for the feedback! (2/2)**
> >
> > ### Section4: On the uncertainty captured by a "ground truth" generative model and adversarial perturbations.
> >
> > We are afraid that there might have been some confusion here. We train a generative model on real data and then generate artificial data with said generative model. In the realm of artificial data, the generative model is the ground truth. Thus, its uncertainty represents the true aleatoric uncertainty of the artificial data with respect to its generating process. We make no claims about the true aleatoric uncertainty of the real dataset / real data generating process. Our goal is simply to set up an artificial environment where we control the data generating process.  Section 4 has been updated to further clarify this.
> >
> > We are not sure we understand your comment regarding the ensemble. Could you please elaborate? -- An ensemble might capture uncertainty in our model’s specification (uncertainty in the weights / architecture). This is known as epistemic uncertainty. We are not sure how it would be relevant to our scenario.
> >
> > With regards to our comment “Approaches that exploit adversarial weaknesses in the BNN will not transfer to the g.t. VAEAC, failing to reduce uncertainty or error”; you could imagine an adversarial attack, which takes in an uncertain input and modifies it such that the uncertainty of our BNN is minimized while not differing from the original input in any meaningful way (this is illustrated in figure 3). Because this attack targets the BNN and not the data generating model, an input constructed in such a way would still seem uncertain to our ground truth generative model and would fare poorly on our metrics.
> >
> > ### Additional Questions:
> >
> > *Projected Gradient descent:* This is a very interesting idea which could lead to smoother optimization. However, it is not trivial what the projection step would be in our case.
> >
> > *Penalty on prediction similarity $d_{y}$:* This penalty can be leveraged to generate counterfactuals which our model believes to belong to a certain class. Which class that is will be scenario dependent. You could imagine a case where you know what the target class is and you would like to find a low uncertainty input which is similar to your original input while being classified correctly. You could also imagine a setting where you want to gain information about a model’s uncertainty while preserving the originally predicted class. Our experiments in the main text do not cover those cases. Thus, we set the prediction similarity loss weight $\lambda_{y}$ to be 0. We experiment with the prediction similarity loss in appendix H.1
> >
> > *Choice of FIDO baseline - use of contiguous set constraints:* We choose FIDO as the method to adapt to uncertainty because it seemed similar to CLUE and produced state of the art results for counterfactual generation for classification. You are correct in that the original FIDO formulation includes a smoothness penalty which encourages adjacent pixels to have the same mask values. This makes sense in the context of high resolution images. We do not consider this penalty, as our experiments focus on tabular data, where there is no local coherence among input dimensions and where counterfactual explanations are most commonly used in practise. MNIST images are only sized (28x28). They are not large enough to warrant spatial coherence constraints.
> >
> > ### Other Suggestions:
> >
> > Thanks, we have incorporated your comments in the new submission
> >
> > *Algorithm 1 nomenclature:* Thanks, we have updated the algorithm to refer to counterfactuals in the sense of uncertainty.
> >
> > *Clarification on step 4 of section 4:* You are right, the g.t. VAEAC is used to measure the density of the counterfactuals under the true generative process of the artificial data $p_{gt}(\bar{x_{c}})$. This is useful because it tells us if the counterfactuals represent plausible input variable settings. The g.t. VAEAC is also used to measure the uncertainty of the counterfactuals through their conditional density $p_{gt}(y|\bar{x_{c}})$. We can use this distribution to calculate the uncertainty associated with $\bar{x_{c}}$ under the true generative process of the artificial data $H_{gt}(y|\bar{x_{c}})$. The expressions necessary to do this are provided in Appendix B.2.  $H_{gt}(y|\bar{x_{c}})$ provides us with a reliable way to see if counterfactual explanations represent low uncertainty configurations. We have further clarified the above in section 4.
> >
> > *Constraining explanations to to small contiguous regions:* We see this as being relevant to methods that are geared towards natural images, as is the case in the papers that you cite. As suggested, we now mention this type of constraint at the end of section 3.

---

### Author Response · Authors · 2020-11-19
**Paper update overview**

We would like to thank all reviewers for their time in reviewing our paper. We are glad the reviewers appreciated the novelty of designing methods aimed at increasing the transparency of uncertainty estimates in ML. We are also happy that the reviewers enjoyed seeing our user study and found its results persuading.

We thank the reviewers for their helpful suggestions. We have incorporated these to improve our paper; **the newly uploaded manuscript contains the following changes**:
* We have modified Section 1 (introduction) to make it clear that the described workflow and Figure 1 are simply an example of a possible application of CLUE, not the main guiding motivation behind CLUE. We also better explain why feature attribution is typically not suitable for uncertain inputs.
* In Section 1, we clarify that our proposed method, CLUE, can be applied to any probabilistic model, not just BNNs. However, this work focuses on BNNs.
* In Section 2.3, we added a comment relating counterfactuals to the broader category of contrastive explanations and cited relevant work suggested by AnonReviewer4.
* In Section 2.3, we added a citation to Bayes-TrEx (https://arxiv.org/abs/2002.10248)
* In Algorithm 1, we updated the notation used to be more general.
* In section 3, we have added a short discussion on the usage of contiguity/smoothness constraints for saliency maps and added relevant citations.
* In Section 4, we go into additional detail on how “ground-truth generative models” are used to evaluate counterfactual examples in our proposed evaluation framework.
* In Section 4, we have added a reference to Appendix I, where we give extensive details on the “ground-truth generative models” used in our evaluation framework.
* In Section 5.2, we clarify that our user study subjects are graduate students in ML, a proxy for ML practitioners.
* **We perform an additional experiment where the typicality of counterfactual explanations is measured through Nearest Neighbour distance to the training set, as suggested by AnonReviewer2**


 We are open to your feedback and additional suggestions.

---

### Comment · ~Sumedha_Singla1 · 2022-02-20
**Add reference**

Thank you for your work. Interesting read. The authors should consider citing [1] as it proposes a generative method for deriving counterfactual explanation, very relevant to the current work.

[1]  Singla, Sumedha, Brian Pollack, Junxiang Chen, and Kayhan Batmanghelich. "Explanation by Progressive Exaggeration." In International Conference on Learning Representations. 2019.

---

### Decision · Program_Chairs · 2021-01-07
**Final Decision**

**Decision:**

Accept (Oral)

**Comment:**

This paper presents an uncertainty quantification method that is conceptually interesting and practical. All reviewers are in consensus regarding the quality and significance of this manuscript.